# *ManiSoft*: Towards Vision-Language Manipulation for Soft Continuum Robotics

**Ziyu Wei** [* 1 2]   **Luting Wang** [* 1 2]   **Chen Gao** [◇ † 1 3]   **Li Wen** [† 1]   **Si Liu** [† 1 2]

Project page: https://buaa-colalab.github.io/ManiSoft

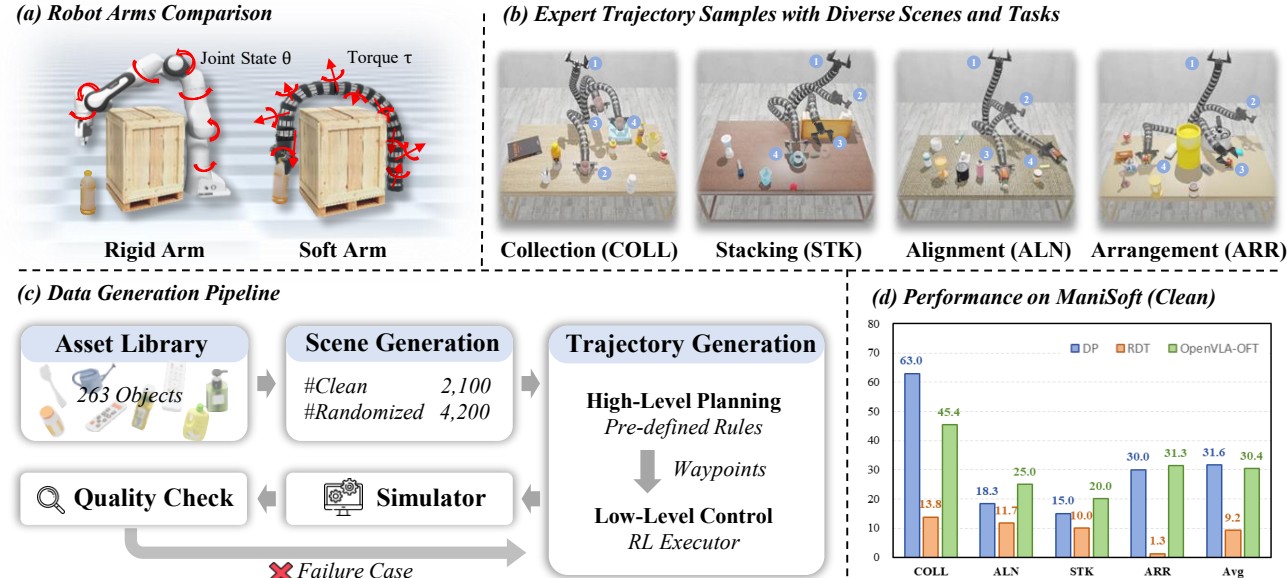

*Figure 1.* (a) Rigid arms operate in a low-dimensional action space and can fail due to limited shape adaptation, whereas soft arms driven by distributed low-level actuation can continuously deform to reach around obstacles. (b) Example expert trajectories for the four *ManiSoft* tasks. (c) The data generation pipeline comprises an asset library, clean and randomized scene generation, hierarchical trajectory generation, simulation, and quality check. (d) Performance of representative policy models on *ManiSoft* in clean scenarios.

## Abstract

Most existing vision-language manipulation research targets rigid robotic arms, whose fixed morphology limits adaptability in cluttered or confined spaces. Soft robotic arms offer an appealing alternative due to their deformability, but confront challenges such as unreliable proprioception and distributed low-level actuation. To investigate these challenges, we introduce *ManiSoft*, a benchmark for vision-language manipulation with soft arms. *ManiSoft* features a tailored simulator that couples realistic soft-body dynamics with contact-rich interactions via an elastic force constraint. On this basis, *ManiSoft* defines four tasks, each highlighting distinct aspects of deformable control, from basic end-effector coordination to obstacle avoidance. To support policy training and evaluation, *ManiSoft* includes an automated pipeline that generates 6,300 diverse scenes and corresponding expert trajectories. To produce high-quality trajectories at scale, we first employ a high-level planner to decompose each task into a sequence of waypoints, followed by a low-level reinforcement learning policy that generates torque commands to track waypoints. Benchmarking three representative policy models shows relatively promising results in clean scenes but substantial per-

* Equal contribution. ◇ Project lead. † Corresponding author. [1]Beihang University [2]Hangzhou Innovation Institute, Beihang University [3]National University of Singapore. Correspondence to: Chen Gao <gaochen.ai@gmail.com>, Li Wen <liwen@buaa.edu.cn>, Si Liu <liusi@buaa.edu.cn>.

*Proceedings of the 43rd International Conference on Machine Learning*, Seoul, South Korea. PMLR 306, 2026. Copyright 2026 by the author(s).

formance drop under randomization. Visualization analysis indicates that failures stem primarily from inaccurate visual estimation of proprioceptive state and limited exploitation of deformability for adaptive obstacle avoiding. We anticipate *ManiSoft* to serve as a valuable testbed, bridging the gap between rigid and soft arms in the context of vision-language manipulation.

## 1. Introduction

Vision-language manipulation (Shao et al., 2025) is a central capability of embodied AI, enabling language-conditioned interaction with the physical world. To date, most benchmarks (Liu et al., 2023; Yu et al., 2021; Mees et al., 2022; Li et al., 2024b; Srivastava et al., 2022) and methods (Chi et al., 2025; Liu et al., 2025b; Kim et al., 2024; 2025) focus on rigid robotic arms, where accurate proprioception and low-dimensional kinematics enable straightforward perception-to-control pipelines. However, rigid morphologies impose fundamental limitations in cluttered or confined environments (Chen et al., 2025b). As illustrated in Figure 1 (a), when obstacles require significant shape adaptation, a rigid arm may fail to reach the target due to its joint constraints.

Soft robotic arms (Xie et al., 2023; Armanini et al., 2023; Majidi, 2014; Hughes et al., 2016; Zhao et al., 2024), built from elastic materials, offer an appealing alternative. Through continuous deformation, soft arms can adapt their geometry to execute policies infeasible for rigid arms. However, these advantages come with major challenges for vision-language manipulation. Unlike rigid arms with reliable joint sensing, soft arms often lack accurate proprioception (Pagliarani et al., 2025), leading to highly complex kinematic control. Therefore, soft arms are typically actuated via low-level commands (e.g., pressures (Liu et al., 2025a), tendon tensions (Walker et al., 2024), or torques (Caasenbrood et al., 2022)) rather than intuitive kinematic targets. Moreover, distributed actuation along the body yields a higher-dimensional and more coupled action space than rigid arms. Together, these factors complicate the generation of stable and coordinated behaviors.

In this work, we introduce *ManiSoft*, a benchmark designed to catalyze vision-language manipulation research for soft arms. *ManiSoft* provides (i) a simulation and rendering stack for soft-arm manipulation, (ii) a suite of language-conditioned tasks with diverse scenes, and (iii) expert demonstration trajectories to support imitation and offline reinforcement learning. This design allows for evaluating existing policy models with minimal modifications, exposing failure modes unique to deformable embodiments.

A central technical challenge lies in simulation. While many soft-body simulators (Huang et al., 2021; Faure et al., 2012; Wei et al., 2025) accurately capture elastic dynamics, they offer limited support for environmental interactions (e.g., contact and friction). In contrast, rigid-body simulators (Geng et al., 2025; Xiang et al., 2020; Szot et al., 2021) excel at modeling interactions but struggle with continuous deformation. To bridge this gap, we integrate a soft-body dynamics simulator (Naughton et al., 2021) with a rigid-body interaction simulator (Todorov et al., 2012) through an elastic force constraint, facilitating contact-rich manipulation with a soft arm. We further provide a Blender-based[1] renderer for generating the visual observations used by policy models.

Built on this stack, *ManiSoft* defines four tasks as illustrated in Figure 1 (b). For each task, we follow the automated pipeline in Figure 1 (c) to generate tabletop scenes and expert trajectories. The asset library contains 263 3D objects, annotated with candidate manipulation poses. We first construct clean (uncluttered) scenes by sampling target objects from the asset library and then create randomized variants by adding obstacles and varying object placements and textures, enabling systematic evaluation under increasing visual and physical complexity.

For each scenario, we generate expert trajectories using a hierarchical mechanism. A high-level planner produces a sequence of waypoints, where each waypoint specifies a 6-DoF end-effector pose. A low-level controller outputs torque commands to drive the soft arm between successive waypoints. In our implementation, the high-level planner uses task-specific rules, and the low-level controller is a reinforcement learning (RL) (Sutton, 1988) policy. This decomposition mitigates the difficulty of directly producing torque sequences and yields stable trajectories for training.

Finally, we benchmark representative policy models on *ManiSoft*. As summarized in Figure 1 (d), existing models can solve a subset of tasks in clean scenes. However, performance drops substantially in randomized settings. Our visualizations and failure-case analysis suggest two bottlenecks: (i) estimating the soft arm's proprioceptive states from visual observations, and (ii) exploiting deformability to plan obstacle-avoiding interaction strategies. We hope *ManiSoft* will serve as a testbed for developing methods that address these challenges.

**Conflict of Interest Disclosure**. We declare that we have no relevant or material financial interests that relate to the research described in this paper.

---

[1]https://www.blender.org/

## 2. Related Works

**Robotic Manipulation Benchmarks.** Progress in vision-language manipulation has been accelerated by benchmarks that standardize tasks, observations, and evaluation protocols for comparing policy models (Shridhar et al., 2020; Ahmed et al., 2020; Qi et al., 2020). RLBench (James et al., 2020) provides a suite of 100 vision-based manipulation tasks for evaluating both learning-based and traditional policy models. The ManiSkill series (Mu et al., 2021; Gu et al., 2023; Tao et al., 2025) emphasizes generalizable manipulation over diverse objects in a full-physics simulator. CALVIN (Mees et al., 2022) targets long-horizon, language-conditioned manipulation, while LIBERO (Liu et al., 2023) studies cross-task transfer in lifelong learning. RoboVerse (Geng et al., 2025) supports evaluation across multiple simulators and robot embodiments. RoboTwin (Mu et al., 2024; Chen et al., 2025a) proposes an automated pipeline for generating diverse dual-arm manipulation scenarios at scale. Despite their breadth, these benchmarks predominantly target rigid arms with low-dimensional kinematics and reliable proprioception, leaving vision-language manipulation for deformable embodiments relatively underexplored. We fill this void by introducing a benchmark tailored to vision-language manipulation with soft arms.

**Vision-Language-Action Models.** Vision-Language-Action (VLA) models have advanced rapidly in recent years. RT-1 (Brohan et al., 2023b) and RT-2 (Brohan et al., 2023a) demonstrate the effectiveness of large-scale training by leveraging multi-robot datasets. DexVLA (Wen et al., 2025) extends this paradigm to enhance efficiency and generalization in long-horizon manipulation. RDT-1B (Liu et al., 2025b) introduces a diffusion-based foundation model for bimanual manipulation. OpenVLA (Kim et al., 2024) presents an open-source framework built upon large language models and pretrained visual encoders, while CogACT (Li et al., 2024a) proposes an action module conditioned on VLM outputs to improve action prediction. More recently, the $\pi$ series (Black et al., 2025b;a; Amin et al., 2025) has demonstrated strong performance via large-scale pretraining followed by reinforcement learning. Despite this progress, prior VLA methods have been predominantly developed and evaluated on rigid arms. Our work provides a comprehensive benchmark and systematic evaluation of representative policy models on soft arms, highlighting unique challenges absent in rigid arms.

**Soft Robotic Arms.** Over the past decade, soft arms have been widely studied and applied in domains such as biomedical engineering (Cianchetti et al., 2018; Rogatinsky et al., 2023), aerospace (Ruiz et al., 2024; Szász et al., 2022), and underwater exploration (Gong et al., 2021; Li

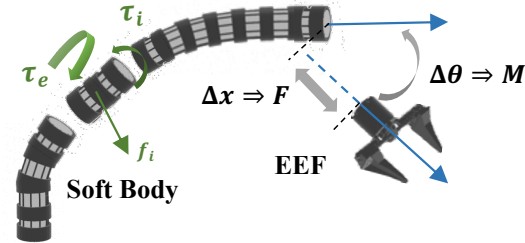

*Figure 2.* **Overview of the soft arm modeling in our Simulator.** The soft body is modeled as a Cosserat rod that moves under the influence of an external torque $\tau_e$. Interaction between soft body and EEF is represented via an an elastic force constraint. Relative displacement $\Delta\mathbf{x}$ or relative rotation $\Delta\theta$ between them induces corresponding restoring forces and torques.

et al., 2023). To address the challenges of modeling and control, learning-based approaches have been extensively explored. Thuruthel et al. (2017) applies trajectory optimization for open-loop predictive control, while Thuruthel et al. (2018) extends this framework using model-based reinforcement learning for closed-loop control. Centurelli et al. (2022) develops a controller based on LSTM and TPRO for dynamic trajectory tracking, both with and without payloads. Soft DAgger (Nazeer et al., 2023) enables sample-efficient imitation learning for soft control. While these efforts advance low-level control of soft arms, high-level vision-language reasoning for manipulation remains largely unaddressed. Our work studies vision-language manipulation with soft arms, which requires jointly reasoning about visual perception, language understanding, environmental interaction, and deformable control.

## 3. The *ManiSoft* Benchmark

We introduce the *ManiSoft* benchmark, designed to support vision-language manipulation with soft robotic arms. It comprises a soft-arm simulator, a collection of diverse tabletop scenes paired with language instructions, and expert demonstration trajectories. To enable scalable data collection, we propose an automated data generation pipeline that integrates procedural scene construction with a hierarchical expert trajectory generation mechanism.

### 3.1. Simulator

While existing soft-body simulators faithfully capture elastic deformation, they often provide limited support for interactions with rigid environments. Conversely, rigid-body simulators excel in modeling contacts and friction but lack native continuous deformation. To bridge this gap for soft robotic manipulation, we develop a hybrid simulator that combines accurate deformable dynamics with robust environmental interactions.

As illustrated in Figure 2, we model the soft arm as two coupled components: a deformable soft body and an end-effector. These components are linked via an elastic force constraint to ensure coordinated yet compliant motion.

The soft body is simulated using Elastica (Naughton et al., 2021), which discretizes the arm into $N$ segments following the Cosserat rod theory (Cosserat & Cosserat, 1909). External actuation torques $\boldsymbol{\tau}_e \in \mathbb{R}^{N \times 3}$ induce axial, shear, bending, and torsional strains along the rod, producing internal forces $\mathbf{f}_i \in \mathbb{R}^{N \times 3}$ and moments $\boldsymbol{\tau}_i \in \mathbb{R}^{N \times 3}$. These forces and torques govern the deformation together. Appendix A provides further details on Cosserat rod theory.

The end-effector and its interactions with the environment are handled by MuJoCo (Todorov et al., 2012), enabling efficient and stable simulation of contact-rich scenarios.

To couple the soft body and the end-effector, we impose an elastic force constraint. The two components are connected by a stretchable and twistable virtual spring with zero rest length. Relative translations $\Delta\mathbf{x} \in \mathbb{R}^3$ and rotations $\Delta\boldsymbol{\theta} \in \mathbb{R}^3$ between the attachment points generate restoring force $\mathbf{F} \in \mathbb{R}^3$ and torque $\mathbf{M} \in \mathbb{R}^3$, computed according to Hooke's law:

$$\mathbf{F} = -k_F \Delta\mathbf{x}, \quad \mathbf{M} = -k_M \Delta\boldsymbol{\theta}, \tag{1}$$

where $k_F, k_M \in \mathbb{R}$ are the translational and rotational stiffness coefficients, respectively. These restoring terms penalize relative motion between the soft body and the end-effector, driving the system toward coordinated motion.

We employ Blender to render the visual observations. Based on the simulated states, Blender produces RGB images of the tabletop scenes from fixed camera viewpoints, including the soft arm, target objects, and surrounding obstacles. Rendering parameters are detailed in Appendix B.

### 3.2. Task Definition

As shown in Figure 1 (b), *ManiSoft* defines four manipulation tasks, each designed to highlight distinct challenges in vision-language manipulation for soft arms. Collecting (**COLL**) involves guiding the soft arm to gather a designated object and deposit it into a container, thereby evaluating fundamental trajectory control and basic end-effector coordination in policy models. Building on this foundation, Alignment (**ALN**) demands precise positioning of the target object to a specified 6-DoF pose, testing the model's capability for fine-grained orientation adjustments. Stacking (**STK**) escalates the challenge by requiring the arm to assemble tableware items from largest to smallest in a stable vertical pile, which assesses precision control during elevated, contact-rich interactions. Finally, Arrangement (**ARR**) requires placing objects according to a specified spatial configuration, thereby demanding integrated visual

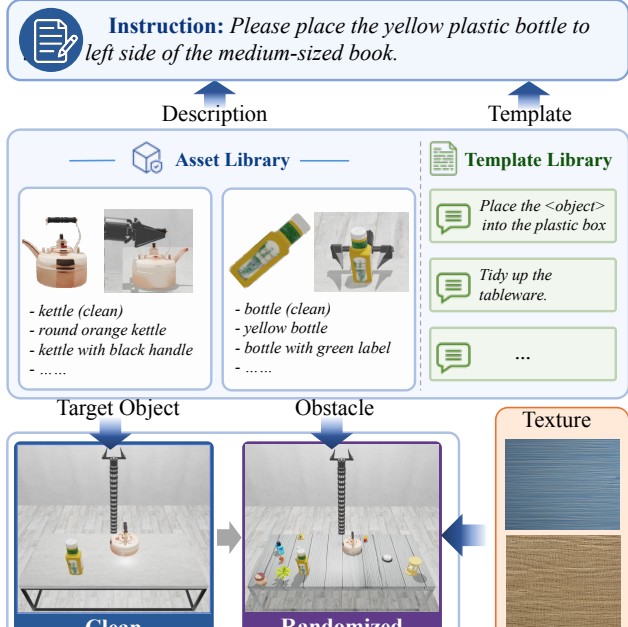

*Figure 3.* **Scene generation in *ManiSoft*.** Objects are sampled from the asset library to create a clean scene, and randomized scenes are generated by injecting objects as obstacles and varying surface textures. Instructions are produced with the descriptions of relevant objects. In the randomized setting, diverse descriptions are leveraged to enhance the linguistic richness.

perception, spatial reasoning, and obstacle avoidance.

Formally, at each time step $t \in \mathbb{N}^+$, given an instruction $\mathbf{L}$ and the current visual observation $\mathbf{V}_t$, the policy model predicts the next action $\mathbf{A}_t = (\boldsymbol{\tau}_e, S)$. Here, $\boldsymbol{\tau}_e$ is the external torques and $S \in \{0, 1\}$ indicates the end-effector state. Upon execution, a new observation $\mathbf{V}_{t+1}$ is rendered, and the policy model proceeds autoregressively until task completion or the maximum horizon $T$ is reached.

Unlike rigid-arm benchmarks, which typically provide proprioceptive state (e.g., joint angles), *ManiSoft* deliberately excludes internal soft-body states to reflect real-world sensing limitations. Policy models must therefore infer the arm's configuration and deformation solely from visual observations, introducing significant challenges in proprioceptive state estimation and deformable strategy planning.

Two evaluation metrics are used: (i) success rate, determined by task-specific criteria, and (ii) efficiency, measured as the number of steps required for completion.

### 3.3. Scene Generation

As depicted in Figure 3, *ManiSoft* adopts a tabletop environment as its core setting. The soft arm is fixed behind the table and centered relative to the workspace, allowing full access to objects distributed across the table sur-

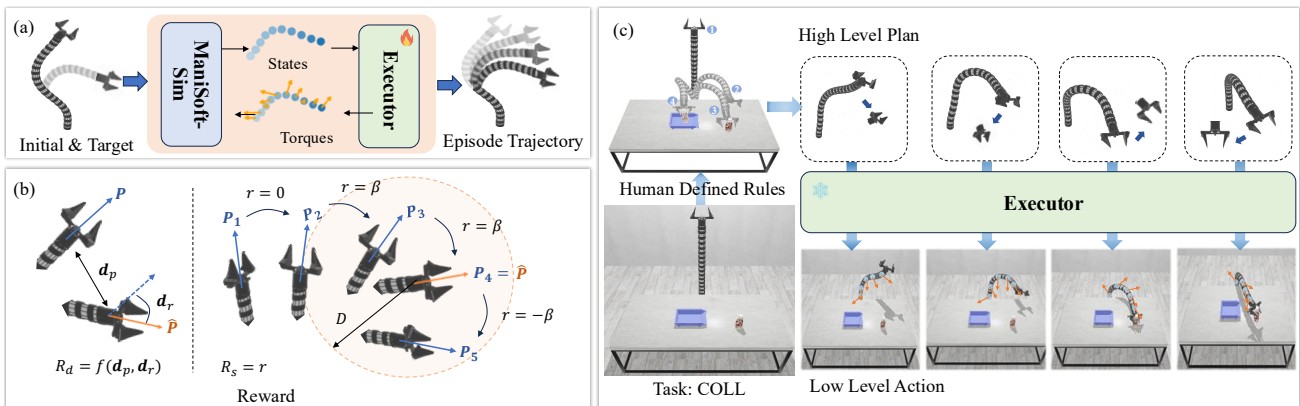

*Figure 4.* **Trajectory generation pipeline in *ManiSoft*.** (a) An executor is trained via RL policy to transform waypoint (6-DoF pose) into torques. (b) RL rewards are designed to balance accuracy and stability, consisting of a pose difference reward $R_d$ negatively correlated with the pose difference, and a stability reward $R_s$ that penalizes or rewards changes in pose difference. (c) Task-specific rules are predefined to produce high-level planning (trajectory waypoints) for each case, which are then converted into low-level actions (torque commands) by the executor to generate complete trajectories.

face. We build our object library by leveraging assets from RoboTwin-OD (Chen et al., 2025a). For each object, we pre-annotate a set of suitable 6-DoF end-effector poses for interaction with the soft arm (e.g., approach, grasp, and lift candidates). These annotations guide the high-level planner during expert trajectory generation. Scenes are procedurally generated by randomly sampling objects from this library and placing them within the workspace.

To support systematic evaluation, each task includes two difficulty levels: clean and randomized. In the clean setting, scenes contain only the task-relevant target objects in fixed layouts and appearances. In the randomized setting, we introduce additional irrelevant objects as obstacles to enhance spatial complexity, and apply scene randomization by sampling diverse textures along with variations in lighting intensity and brightness. This yields diverse spatial arrangements and visual appearances across episodes, stressing generalization in perception and planning.

Language instructions are generated in a controlled manner to ensure diversity, semantic accuracy, and consistency. Direct LLM sampling often produces variable phrasing or minor hallucinations; therefore, we first generate candidate instructions via GPT, manually curate and refine them into a template library, then instantiate templates by filling in object attributes (e.g., color, shape, material). In clean scenes, each object receives a single canonical description. In randomized scenes, objects are paired with multiple attribute-aware descriptions to reflect visual variability (e.g., a bottle may be referred to as "yellow bottle", "bottle with green cap", or "tall plastic bottle").

### 3.4. Trajectory Generation.

Given the procedurally generated scenes for each task, we produce expert trajectories using a hierarchical mechanism, as illustrated in Figure 4. At the high level, a task-specific rule-based planner generates a sequence of waypoints, where each waypoint defines a desired 6-DoF end-effector pose in $SE(3)$. These waypoints encode semantically intermediate configurations (e.g., approach, grasp, retract) tailored to the task, avoiding the need for direct torque-sequence planning over long horizons.

At the low level, an RL-trained executor drives the soft arm from its current configuration to each successive waypoint using torque actuation. At each timestep $t$, the executor receives the following inputs: (i) the target end-effector pose $\hat{P} \in SE(3)$, (ii) proprioceptive states including positions and velocities of selected segments along the arm, and (iii) the current end-effector pose $P \in SE(3)$, and outputs torque commands.

The executor is trained with a dense reward function that encourages precise reaching of the target waypoint while promoting stable convergence and penalizing excessive deformation or oscillation. We measure the pose discrepancy using the standard $SE(3)$ logarithm map:

$$[\mathbf{d}_p, \mathbf{d}_r] = \log(P^{-1}\hat{P}), \quad d = \|\mathbf{d}_p\|_2 + \alpha\|\mathbf{d}_r\|_2, \quad (2)$$

where $\mathbf{d}_p, \mathbf{d}_r \in \mathbb{R}^3$ represent the position and axis-angle rotation differences, respectively. The scalar $\alpha > 0$ is tuned to balance the contributions of translation and rotation.

We define the total reward $R$ as the sum of two terms: a pose difference term $R_d$ and a stability term $R_s$. The pose difference reward $R_d$ is adapted from Elastica-RL-Control (Naughton et al., 2021), using the pose distance

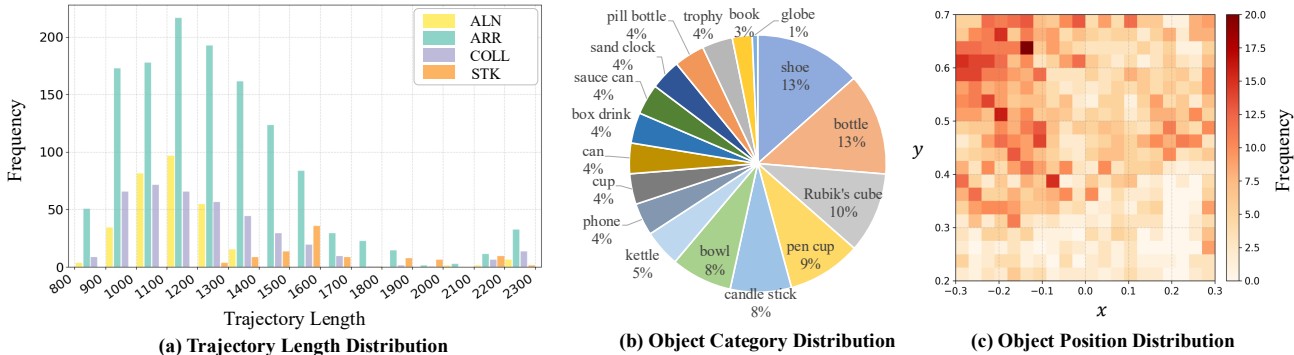

**(a) Trajectory Length Distribution**  **(b) Object Category Distribution**  **(c) Object Position Distribution**

*Figure 5.* **Statistical analysis of the *ManiSoft* Benchmark.** (a) Distribution of trajectory lengths. Tasks in *ManiSoft* generally involve long trajectories, with the STK task exhibiting notably longer trajectories than the others. (b) Frequency distribution of target object categories, highlighting the diversity of manipulable objects in *ManiSoft*. (c) Spatial distribution of initial target object positions on the tabletop, showing that graspable targets are broadly and evenly distributed across the workspace.

$d$ instead of Euclidean distance:

$$R_d = -d + k_1 \mathbb{1}_{\{d<d_1\}} + k_2 \mathbb{1}_{\{d<d_2\}}. \qquad (3)$$

The stability reward $R_s$ provides a signal based on the rate of change of $d$ when the end-effector is close to the target pose, encouraging smooth and stable convergence:

$$R_s = \begin{cases} -\operatorname{sgn}\left(\dfrac{\partial d}{\partial t}\right) \cdot \beta, & d \le D, \\ 0, & d > D, \end{cases} \qquad (4)$$

where $\beta > 0$ is a scaling factor that controls the strength of the stability incentive.

Once trained, the executor is used to roll out complete trajectories by sequentially tracking the high-level waypoints. This hierarchical decomposition produces stable, collision-free demonstrations across a wide range of scenes. It significantly simplifies downstream policy learning compared to training directly on raw torque actions.

Empirically, the executor reaches a success rate of $54\%$ on 100 random samples. We further investigate the effect of the stability reward $R_s$ on control stability. As shown in Figure 6, the model trained with $R_s$ exhibits noticeably smaller fluctuations in pose difference compared to the model trained without $R_s$. We also perform an ablation study on different parameter settings of $R_s$. The variance of the pose difference between the end-effector and the target is used to quantify control stability. Based on this metric, we select the best-performing set of parameters, $\beta = 1$, $D = 0.3$, as reported in Table 1. Additional ablation studies are provided in Appendix C.

### 3.5. Data Statistic

*ManiSoft* contains 6,300 scene–trajectory pairs, comprising 2,100 clean scenes and 4,200 randomized scenes, with an

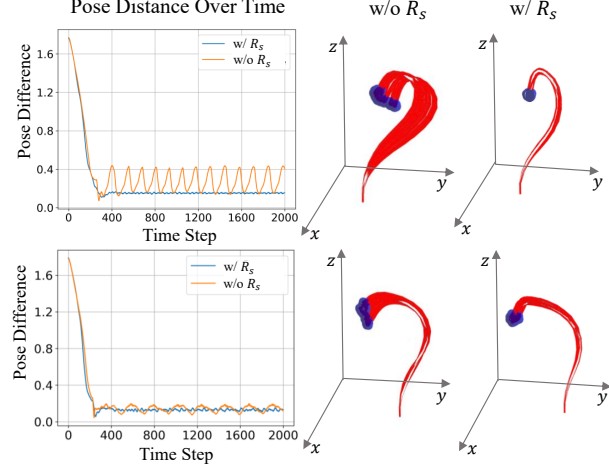

*Figure 6.* **Visualization of executor trained w/ and w/o the stability reward $R_s$.** (Left) The pose difference between the end-effector and the target pose over time. (Right) The soft robotic arm's trajectory shadows during the final 1000 simulation steps. The red line represents the soft body, and the blue circle indicates the end-effector.

average of 40 language instructions per scene. The dataset is split into training and testing sets with a ratio of $4:1$.

Owing to the high precision of torque-based control, trajectories in *ManiSoft* are relatively long, with an average length of 1,272 steps; the distribution of trajectory lengths is shown in Figure 5 (a). *ManiSoft* features a rich variety of objects, including 109 manipulable objects across 17 categories, as illustrated in Figure 5 (b), and 154 obstacles spanning 35 categories. Figure 5 (c) visualizes the initial positions of target objects (i.e. grasp poses) across all scenes as a heatmap over the tabletop, demonstrating a wide distribution of grasp positions across the entire table.

Examples of the generated data are shown in Figure 7.

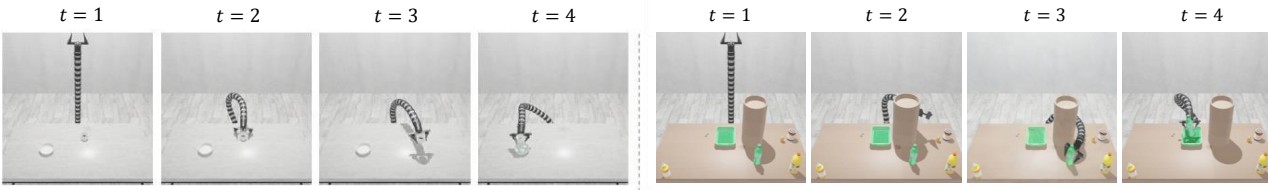

**Instruction:** *Arrange the tableware in stacks sorted by size.*

**Instruction:** *Position the green bottle inside the basket.*

*Figure 7.* **Visualization of the *ManiSoft*.** The left example is a clean scene, while the right is a randomized scene.

*Table 1.* Control stability under different parameters of $R_s$. Specifically, $\beta = 0$ indicates the absence of $R_s$.

| $D$ \ $\beta$ | 0 | 0.5 | 1 | 1.5 |
|---|---|---|---|---|
| 0.05 | 0.176 | 0.157 | 0.074 | 0.121 |
| 0.1 | 0.176 | 0.149 | 0.153 | 0.071 |
| 0.2 | 0.176 | 0.070 | 0.135 | 0.064 |
| 0.3 | 0.176 | 0.145 | **0.053** | 0.091 |
| Average | 0.176 | 0.130 | 0.104 | 0.087 |

More visualizations are provided in Appendix F.

## 4. Experiments

We evaluate three representative models on *ManiSoft*: Diffusion Policy (DP) (Chi et al., 2025), RDT (Liu et al., 2025b), and OpenVLA-OFT (Kim et al., 2025). DP and RDT are trained from scratch, while OpenVLA-OFT is fine-tuned with LoRA. Implementation details are provided in Appendix D.

### 4.1. Main Results

Table 2 shows the performance of different models on each task under clean and randomized settings. Overall, DP and OpenVLA-OFT achieve substantially better performance than RDT. Specifically, DP attains an average success rate of 31.6% with a mean execution length of 520 steps, while OpenVLA-OFT achieves a comparable success rate of 30.4% with an average of 527 steps. In contrast, RDT performs significantly worse, with an average success rate of only 9.2% despite requiring 496 steps on average. This performance gap is likely attributable to differences in model capacity. RDT contains approximately 1B parameters, whereas DP and OpenVLA-OFT each have around 400M parameters. As a result, RDT is more prone to overfitting the training data, leading to inferior generalization on the testing set. Across all three models, the best performance is consistently observed on the COLL task, which requires less precise orientation perception and spatial reasoning compared to the other tasks.

Under the clean setting, DP achieves higher accuracy than OpenVLA-OFT on the COLL task, exceeding it by 17.6%.

However, OpenVLA-OFT outperforms DP on all remaining tasks. Specifically, it achieves improvements of 6.7%, 5.0%, and 1.3% on the ALN, STK, and ARR tasks, respectively. These results suggest that DP is more effective for simpler tasks, whereas OpenVLA-OFT exhibits stronger reasoning and generalization capabilities in more complex scenarios, likely benefiting from its pretrained weights.

Under the randomized setting, all models experience a decrease in success rate. Specifically, DP exhibits the largest performance drop, with its success rate decreasing by 29.4%, while RDT and OpenVLA-OFT show more moderate declines of 7.6% and 3.4%, respectively. This indicates that the introduction of obstacles, together with variations in language instructions and scene configurations, substantially increases task difficulty. Notably, unlike in the clean setting, OpenVLA-OFT consistently outperforms DP under randomization, achieving an average improvement of 13.1%. This suggests that OpenVLA-OFT maintains stronger generalization performance in the presence of environmental and instruction-level variations.

Table 3 illustrates the performance across different object categories. The Rubik's Cube consistently yields the highest success rates among all objects. In the clean setting, its success rate exceeds that of the other categories by 5%-30% across methods. Notably, under the randomized setting, OpenVLA-OFT achieves a 15.0% success rate on the Rubik's Cube, which is twice that of others. In contrast, the shoe is the most challenging object. In the clean setting, its success rate is 5%–30% lower than other categories across models. This gap becomes more pronounced under randomization, where success rates drop below 10% for all methods. These results indicate that while object geometry strongly affects absolute task difficulty, different models exhibit consistent relative performance trends across object categories. More results are provided in Appendix E.

### 4.2. Analysis

By visualizing rollouts from the evaluated policy models, we identify three failure modes.

**Proprioceptive state ambiguity.** Reliable torque control depends on precise proprioceptive state estimation. Soft-

*Table 2.* The results of the policy under the clean and randomized settings in *ManiSoft*. ACC denotes the task success rate on the eval set, and #Steps represents the number of inference steps required by the model to complete each task. OFT denotes OpenVLA-OFT.

| Method | COLL | | ALN | | STK | | ARR | | Average | |
|---|---|---|---|---|---|---|---|---|---|---|
| | ACC(%) | #Steps | ACC(%) | #Steps | ACC(%) | #Steps | ACC(%) | #Steps | ACC(%) | #Steps |
| *Clean* | | | | | | | | | | |
| DP (Chi et al., 2025) | **63.0** | 547 | 18.3 | **442** | 15.0 | 517 | 30.0 | 573 | **31.6** | 520 |
| RDT (Liu et al., 2025b) | 13.8 | **509** | 11.7 | 463 | 10.0 | 803 | 1.3 | **210** | 9.2 | **496** |
| OpenVLA-OFT (Kim et al., 2025) | 45.4 | 565 | **25.0** | 472 | **20.0** | **492** | **31.3** | 578 | 30.4 | 527 |
| *Randomized* | | | | | | | | | | |
| DP (Chi et al., 2025) | 3.8 | **521** | 1.7 | **324** | 2.5 | 818 | 0.6 | 790 | 2.2 | 613 |
| RDT (Liu et al., 2025b) | 1.2 | 487 | 4.2 | 379 | 0.0 | - | 1.3 | **238** | 1.6 | **368** |
| OpenVLA-OFT (Kim et al., 2025) | **32.7** | 601 | **26.7** | 489 | **35.0** | 563 | **13.7** | 563 | **27.0** | 554 |

*Table 3.* The results of the policy on each manipulable objects in the ARR task.

| Method | Rubik's Cube | | Bottle | | Pen Cup | | Shoe | | Average | |
|---|---|---|---|---|---|---|---|---|---|---|
| | ACC(%) | #Steps | ACC(%) | #Steps | ACC(%) | #Steps | ACC(%) | #Steps | ACC(%) | #Steps |
| *Clean* | | | | | | | | | | |
| DP (Chi et al., 2025) | **50.0** | 705 | 25.0 | 454 | 25.0 | 602 | 20.0 | 534 | 30.0 | 573 |
| RDT (Liu et al., 2025b) | 5.0 | 210 | 0.0 | - | 0.0 | - | 0.0 | - | 1.3 | 210 |
| OpenVLA-OFT (Kim et al., 2025) | 40.0 | 667 | **30.0** | 430 | **35.0** | 525 | **20.0** | 690 | **31.3** | 578 |
| *Randomized* | | | | | | | | | | |
| DP (Chi et al., 2025) | 0.0 | - | 0.0 | 324 | - | 818 | 2.5 | 790 | 0.6 | 790 |
| RDT (Liu et al., 2025b) | 0.0 | - | 2.5 | 174 | 2.5 | 302 | 0.0 | - | 1.3 | 238 |
| OpenVLA-OFT (Kim et al., 2025) | **15.0** | 728 | **7.5** | 482 | **25.0** | 507 | **7.5** | 536 | **13.7** | 563 |

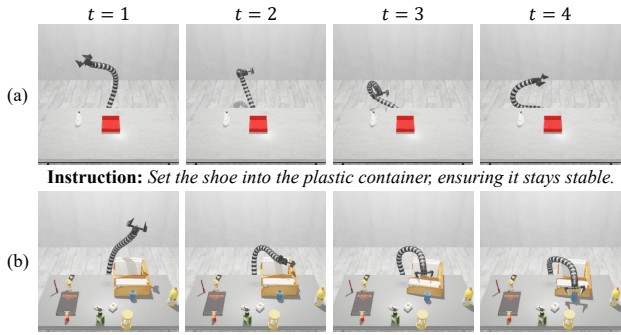

**Instruction:** *Set the shoe into the plastic container, ensuring it stays stable.*

**Instruction:** *Put the blue pencup with rounded edges immediately to the left of the rectangular book.*

*Figure 8.* Visualization of typical failure cases of OpenVLA-OFT in *ManiSoft*. (a) The robot exhibits unexpected torsion and internal forces, resulting in inaccurate action prediction. (b) The robot fails to reach behind the obstacle.

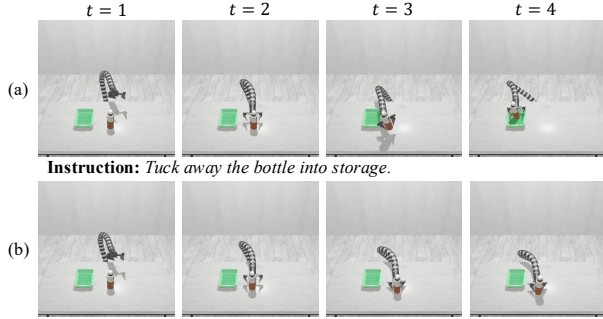

**Instruction:** *Tuck away the bottle into storage.*

**Instruction:** Tuck away the bottle into storage.

*Figure 9.* Comparison of DP and OpenVLA-OFT on the same task: (a) DP successfully completes the task; (b) OpenVLA-OFT exhibits the "stop-moving" behavior.

drifts laterally and ultimately fails to reach the target object.

arm deformation induces internal torques that must be actively compensated. Only the residual torque can drive the arm toward a desired pose. When the compensation term dominates, small state-estimation errors can overwhelm this residual, yielding unreliable control. As shown in Figure 8 (a), the target object lies close to the arm base, requiring a large bend to reach it. This deformation induces substantial internal torques. The policy model fails to compensate for these loads, leaving insufficient residual control to stabilize the motion. Consequently, the end-effector

**Challenges in leveraging soft arm compliance.** Compared to rigid arms, soft arms offer advantages in flexibility, allowing them to adapt their shape to the environment and reach behind obstacles. However, as illustrated in Figure 8 (b), rather than adapting its shape to reach behind the obstacle, the policy model extends the soft arm directly toward the target object, resulting in collisions with the obstacle. This suggests that the policy model has not effectively utilized the soft-specific capabilities, such as shape adaptation and passive compliance. Increasing the proportion of

obstacle-specific expert data or incorporating physical priors during training may help to mitigate this limitation.

**Stop-Moving Behavior.** When comparing DP and OpenVLA-OFT, we observe that OpenVLA-OFT can exhibit a "stop-moving" behavior after grasp completion, where the robot remains stationary and fails to initiate subsequent actions. This behavior is likely caused by subtle visual changes during grasping, which induce a feedback loop that suppresses further action generation. As shown in Figure 9, OpenVLA-OFT often stops moving after a successful grasp, whereas DP rarely encounters this issue. This helps explain why OpenVLA-OFT achieves a lower success rate than DP on the simpler COLL task (45.4% vs. 63.0%) and requires longer execution lengths (565 vs. 547 steps, Table 2). Overall, this highlights a key distinction between diffusion-based and deterministic policies: the stochasticity in diffusion-based policies enables escaping such feedback loops, while deterministic policies are more prone to repetitive behavior.

## Conclusion

We introduced *ManiSoft*, a benchmark for vision-language manipulation with soft arms. *ManiSoft* features a tailored simulator that couples soft-body dynamics with interactions via an elastic force constraint. Four tasks are designed to highlight distinct challenges in deformable control. An automated pipeline generates 6,300 diverse scenes and corresponding expert trajectories. Quality of the trajectories is ensured through a hierarchical mechanism that combines waypoint decomposition with RL-based torque control. Benchmarking representative policy models shows relatively promising performance in clean scenes but marked degradation under randomization. Failures are primarily attributed to inaccurate visual estimation of proprioceptive state and under-exploitation of deformability.

## Limitations

This work represents an initial step toward benchmarking vision-language manipulation for soft robots. The current setup mainly focuses on a subset of actuation mechanisms and relatively simple tabletop scenarios, and does not yet cover more diverse actuation types or more dynamic, long-horizon tasks. In addition, sim-to-real consistency and physical validation could be further strengthened. These aspects leave room for future improvements in terms of coverage, realism, and evaluation diversity.

## Impact Statement

This work can support the development of safer and more human-friendly robotic systems. By providing a benchmark for vision-language manipulation with soft robots, ManiSoft may facilitate research on compliant interaction, with potential applications in service robotics and assistive or medical settings. Overall, it contributes toward more adaptable and accessible robotic technologies.

## Acknowledgements

This research is supported in part by the Key Research Program of Hangzhou (No. 2025SZD1A56), the National Natural Science Foundation of China (No. 62461160308, U23B2010, 62576024), the Beijing Natural Science Foundation (No. L231011), the Fundamental Research Funds for the Central Universities (No. 501RCQD2025141003), BeiHang GanWei Project (No. 502GWXM2024141001), the National Science Foundation Support Projects (No. 62425303), and the National Key R&D Program of China (No. 2024YFb4707300).

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

## Supplementary Material

## A. Cosserat Rod Theory

In Cosserat Rod Theory (Cosserat & Cosserat, 1909), the elastic rod with $L_0$ and radius $r_0$ is presented as a Cosserat rod composed of $N$ discrete element rods, each of length $L_0/N$ and radius $r_0$. For each element rod, we describe its position in the global frame by $\bar{\mathbf{x}}(s,t) \in \mathbb{R}^3$, and its rotation in the global frame is represented by the rotation matrix $\mathbf{Q}(s,t) = \{\bar{\mathbf{d}}_1, \bar{\mathbf{d}}_2, \bar{\mathbf{d}}_3\}^{-1}$, where $\mathbf{Q}(s,t)$ also defines the transformation between the global and local frames. Specifically, for any vector $\mathbf{v}$ in the local frame and $\bar{\mathbf{v}}$ in the global frame, we have $\mathbf{v} = \mathbf{Q}\bar{\mathbf{v}}$ and $\bar{\mathbf{v}} = \mathbf{Q}^T\mathbf{v}$. Here, $s = L \cdot i/N$ denotes the position of the $i$-th element rod in the material coordinate, $t$ represents time, and $\delta s = L/N$ is the length of each element rod. As $N \to \infty$, $s$ becomes continuous, and $\delta s \to ds$. Subsequent derivations will be carried out in the continuous case.

The normal strain of the rod is described by the stretch factor $e = ds/d\hat{s}$, where $d\hat{s} = L_0/N$, and $N \to \infty$ represents the original length of the element rod. The shear strain of the rod is described by the shear vector in the local frame, $\sigma = \mathbf{Q}(\bar{\mathbf{x}}_s - \bar{\mathbf{d}}_3)$, where $\bar{\mathbf{x}}_s = \partial_s \bar{\mathbf{x}}$ is the centerline tangent in the global frame. At this point, we have the translational velocity $\bar{\mathbf{x}} = \partial_t \bar{\mathbf{x}}$ and the curvature vector $\kappa$ satisfies $\partial_s \mathbf{d}_j = \kappa \times \mathbf{d}_j$, which describes the rate of change of rotation along the material coordinate. The angular velocity $\omega$ satisfies $\omega = \partial_t \mathbf{d}_j = \omega \times \mathbf{d}_j$, which describes the rate of change of rotation over time.

Given the bending $\mathbf{B}$ and shearing $\mathbf{S}$ stiffness matrices, the second area moment of inertia $\mathbf{I}$, the cross-sectional area $A$, and the mass per unit length $\rho$, the dynamics of the Cosserat rod can then be written based on the momentum and angular momentum theorems as follows:

$$\rho A \cdot \partial_t \bar{\mathbf{v}} = \partial_s \left( \frac{\mathbf{Q}^T \mathbf{S} \sigma}{e} \right) + e\bar{\mathbf{f}}, \tag{5}$$

$$\frac{\rho \mathbf{I}}{e} \cdot \partial_t \omega = \partial_s \left( \frac{\mathbf{B}\kappa}{e^3} \right) + \frac{\kappa \times \mathbf{B}\kappa}{e^3} + \left( \mathbf{Q}\frac{\bar{\mathbf{x}}_s}{e} \times \mathbf{S}\sigma \right)$$

$$+ \left( \rho \mathbf{I} \cdot \frac{\omega}{e} \right) \times \omega + \frac{\rho \mathbf{I}\omega}{e^2} \cdot \partial_t e + e\boldsymbol{\tau}. \tag{6}$$

Where $\bar{\mathbf{f}}$ is the force density in the global frame for the Cosserat rod, and $\boldsymbol{\tau}$ is the torque density. In the discrete case, they represent the force and torque acting on each element rod. We propose a simulation framework for soft robotic arms that captures both their deformable dynamics and interactions with the environment. As shown in Figure 2, the arm is modeled as two coupled components: a deformable soft body and an end-effector, connected to allow coordinated motion.

*Table 4.* Key Parameters in the *ManiSoft* Simulator

| Parameter | Value |
|---|---|
| ***Simulator*** | |
| $k_F$ | $0.1\ N/m$ |
| $k_M$ | $10\ N \cdot m/rad$ |
| Simulation timestep | $0.0002\ s$ |
| Control Frequency | $714\ Hz$ |
| ***Soft Arm*** | |
| Length | $1\ m$ |
| Radius | $0.05\ m$ |
| Density | $1000\ kg/m^3$ |
| Poisson's ratio | $0.5$ |
| Young's modulus | $1.0 \times 10^7\ Pa$ |
| ***Render*** | |
| Resolution | $514 \times 514$ |
| Camera Position | $(0, 1.6, 1.6)m$ |
| Camera FOV | $60$ |

## B. Details for *ManiSoft* Benchmark

In the simulation, choosing appropriate values for $k_F$ and $k_M$ is crucial for maintaining both numerical stability and physical realism. If the coefficients are too small, positional and orientational discrepancies may persist, leading to separation between the components. On the other hand, excessively large coefficients can result in overcorrection, causing oscillations or even numerical instability. By carefully tuning the evolution of the elastic constraint, *ManiSoft*-Sim ensures stable, physically consistent coupling between the soft body and the EEF, enabling a coherent simulation of soft robotic manipulation.

Specifically, the parameters of the simulator used in our experiments are listed in the Table 4.

In the *ManiSoft* benchmark, we set the maximum execution horizon to $T = 1500$ steps.

## C. Details for Executor Training

For training the executor, we adopt an MLP-based policy network and employ SAC (Haarnoja et al., 2018), for reinforcement learning. We use a learning rate of $3 \times 10^{-4}$ and a batch size of 256. The model is trained using a total of 160M samples.

We perform training and evaluation with different parameters in the reward function.

First, we examine the success rate using a reward that includes only $R_d$ ($\beta = 0$) across various parameter settings. A case is deemed successful once the pose difference between the end-effector and the target drops below a predefined threshold. In Elastica-RL-Control (Naughton et al., 2021), the parameters are set as $k_1 = 0.5$, $k_2 = 1.5$, $d_1 = 0.1$, and $d_2 = 0.05$. We adopt the same values for $k_1$ and $k_2$. For $d_1$ and $d_2$, since we replace the original Euclidean distance with pose difference, the scale of $d$ changes. To maintain the original ratio between $d_1$ and $d_2$, we scale them proportionally, setting $d_1 = 0.1/\lambda$ and $d_2 = 0.05/\lambda$. The model is trained under different $\lambda$ and $\alpha$ configurations. For each setting, we train on 20M samples. During evaluation, a case is considered successful if $d_p < 0.03$ and $d_r < 0.3$. We randomly sample 100 cases to evaluate the success rate, as summarized in Tab. 6. We found that despite changing $d$ from Euclidean distance to pose difference, the best performance was still achieved when $d_1$ and $d_2$ remained unchanged, *i.e.*, when $\lambda = 1$. With $\lambda$ fixed at 1, we trained on 80M samples with different $\alpha$ values, as shown in Table 6. The best performance was achieved with $\alpha = 0.2$.

Based on the model with the highest success rate, we add $R_s$ and perform post-training on 80M samples. We then compare the stability performance under different parameter settings, as shown in Table 1 of the main text.

Fig. 10 shows the visualization of the trained executor controlling the soft robotic arm to move to the target pose.

## D. Implement Details for Baselines

We train and evaluate the three baselines separately on each of the four tasks. All models are trained on 8 RTX 4090 GPUs.

### D.1. Clean Setting.

For the cleaning setting, we adopt the following training configuration.

**DP (Chi et al., 2025).** We set the batch size to 64 and the learning rate to $1 \times 10^{-4}$. The model is trained for $120,000$ iterations on the COLL task and $60,000$ iterations on each of the other three tasks, using a linear learning rate decay schedule. Since DP does not inherently support language understanding, we employ BERT as the text encoder. The resulting text embeddings are combined with image embeddings to guide action generation.

**RDT (Liu et al., 2025b)** We use a batch size of 32 and a learning rate of $1 \times 10^{-4}$, while keeping the text encoder and vision encoder frozen. The model is trained for $60,000$ iterations on the COLL task and $30,000$ iterations on the remaining tasks, with a cosine learning rate decay schedule.

**OpenVLA-OFT (Kim et al., 2025).** We finetune the model using LoRA based on the official pretrained weights, with a batch size of 4 and a learning rate of $5 \times 10^{-4}$. The model is trained for $60,000$ iterations on the COLL task and $30,000$ iterations on the other three tasks, following a cosine learning rate decay schedule.

*Table 5.* Success rate (%) on 100 samples under different parameters of $R_d$.

| $\alpha$ \ $\lambda$ | 0.4 | 0.6 | 0.8 | 1.0 | 1.2 | 1.4 |
|---|---|---|---|---|---|---|
| 0.02 | 6.0 | 6.0 | 3.0 | 6.0 | 5.0 | 5.0 |
| 0.04 | 6.0 | 6.0 | 16.0 | 19.0 | 0.0 | 2.0 |
| 0.05 | 2.0 | 2.0 | 11.0 | 8.0 | 11.0 | 4.0 |
| 0.10 | 2.0 | 11.0 | 16.0 | 17.0 | 13.0 | 20.0 |
| 0.15 | 3.0 | 4.0 | 8.0 | 19.0 | 9.0 | 16.0 |
| 0.20 | 7.0 | 15.0 | 20.0 | 6.0 | 6.0 | 3.0 |
| Avg. | 4.7 | 7.3 | 12.3 | **12.5** | 7.0 | 8.3 |

*Table 6.* Success rate on 100 samples under difference values of $\alpha$ with $\lambda = 1$.

| $\alpha$ | 0.04 | 0.15 | 0.20 | 0.40 |
|---|---|---|---|---|
| Success Rate (%) | 16.0 | 31.0 | **33.0** | 23.0 |

## D.2. Randomized Setting.

For the randomized setting, we finetune the model initialized from the clean setting checkpoint, training for $20,000$ iterations on COLL and $10,000$ iterations on each of the other three tasks, while keeping all other training configurations unchanged.

## E. Results on Each Category

In COLL, ALN, and ARR, multiple categories of manipulable objects are included. Tab. 7 and Tab. 8 present the results for different categories of manipulable objects in COLL and ALN, respectively.

For the COLL task, DP outperforms both RDT and OpenVLA-OFT (by $17.6\%$ and $49.2\%$ respectively) in the clean setting, while in the randomized setting, OpenVLA-OFT performs better than DP by $28.9\%$. Regarding the number of inference steps, DP performs better than OpenVLA-OFT in both settings (by $18$ on clean and $80$ on randomized). This is due to the stop-moving phenomenon in OpenVLA-OFT, which leads to an increase in inference steps. Although RDT requires fewer inference steps, it completes fewer tasks overall, and the tasks it does complete are simpler, requiring fewer execution steps. This does not accurately reflect its overall performance.

The comparison of success rates across different objects reveals that, compared to the candle stick ($100\%$ success rate on DP) and the can ($85\%$), the shoe ($35\%$) and the sand clock ($35\%$) are more difficult to grasp. This is because they require a fixed grasping direction or have relatively large volumes.

A similar trend is observed in the ALN task, where OpenVLA-OFT achieves a higher success rate than DP (by $6.7\%$ on clean and $25\%$ on randomized), but requires more inference steps (by $30$ on clean and $65$ on randomized). In the clean setting, for the same object such as bottle, the success rate in the COLL task is higher than in the ALN task, indicating that the ALN task is relatively more challenging.

## F. More Visualizations

In Figure 11, Figure 12, Figure 13 and Figure 14, we present more visualizations of the four tasks.

*Table 7.* The results of the policy on each manipulable objects in the COLL task. OFT denotes OpenVLA-OFT

| Category | Clean | | | | | | Randomized | | | | | |
|---|---|---|---|---|---|---|---|---|---|---|---|---|
| | DP (Chi et al., 2025) | | RDT (Liu et al., 2025b) | | OFT (Kim et al., 2025) | | DP (Chi et al., 2025) | | RDT (Liu et al., 2025b) | | OFT (Kim et al., 2025) | |
| | ACC(%) | #Steps | ACC(%) | #Steps | ACC(%) | #Steps | ACC(%) | #Steps | ACC(%) | #Steps | ACC(%) | #Steps |
| Bottle | **70.0** | 543 | 15.0 | **444** | 50.0 | 530 | 5.0 | 749 | 0.0 | - | **45.0** | 569 |
| Pill Bottle | **75.0** | 545 | 10.0 | **420** | 55.0 | 520 | 10.0 | 349 | 5.0 | 113 | **40.0** | 599 |
| Can | **100.0** | 535 | 35.0 | 550 | 60.0 | 597 | 10.0 | **515** | 0.0 | - | **45.0** | 663 |
| Cup | **65.0** | 502 | 15.0 | **442** | 60.0 | 529 | 5.0 | 603 | 5.0 | 516 | **30.0** | 591 |
| Sand Clock | 35.0 | 538 | 15.0 | 588 | **35.0** | **473** | 0.0 | - | 0.0 | - | **15.0** | **454** |
| Shoe | **35.0** | 516 | 20.0 | **480** | 20.0 | 556 | 0.0 | - | 0.0 | - | **20.0** | **580** |
| Candle Stick | **85.0** | **502** | 15.0 | 619 | 60.0 | 530 | 0.0 | - | 0.0 | - | **40.0** | **545** |
| Box Drink | **65.0** | **511** | 15.0 | 530 | 35.0 | 675 | 10.0 | **287** | 5.0 | 833 | **35.0** | 613 |
| Kettle | **50.0** | 634 | 0.0 | - | 45.0 | **602** | 0.0 | - | 0.0 | - | **35.0** | 593 |
| Pen Cup | **80.0** | 489 | 5.0 | **484** | 60.0 | 490 | 5.0 | **455** | 0.0 | - | **40.0** | 735 |
| Sauce Can | **70.0** | 557 | 20.0 | **496** | 40.0 | 532 | 0.0 | - | 0.0 | - | **20.0** | **513** |
| Rubik's Cube | **50.0** | 547 | 10.0 | 744 | 35.0 | 667 | 0.0 | - | 0.0 | - | **30.0** | **708** |
| Trophy | **45.0** | 689 | 5.0 | **309** | 35.0 | 641 | 5.0 | 689 | 0.0 | - | **30.0** | 652 |
| Average | **63.0** | 547 | 13.8 | **509** | 45.4 | 565 | 3.8 | 521 | 1.2 | **487** | 32.7 | 601 |

*Table 8.* The results of the policy on each manipulable objects in the ALN task.

| Setting | Method | Bottle | | Shoe | | Candle Stick | | Average | |
|---|---|---|---|---|---|---|---|---|---|
| | | ACC(%) | #Steps | ACC(%) | #Steps | ACC(%) | #Steps | ACC(%) | #Steps |
| Clean | DP (Chi et al., 2025) | 5.0 | 391 | 5.0 | **519** | **45.0** | 417 | 18.3 | **442** |
| | RDT (Liu et al., 2025b) | 5.0 | 538 | 20.0 | 535 | 10.0 | **316** | 11.7 | 463 |
| | OpenVLA-OFT (Kim et al., 2025) | **15.0** | **370** | **25.0** | 597 | 35.0 | 449 | **25.0** | 472 |
| Randomized | DP (Chi et al., 2025) | 2.5 | **371** | 2.5 | **278** | 0.0 | - | 1.7 | **324** |
| | RDT (Liu et al., 2025b) | 0.0 | - | 7.5 | 498 | 5.0 | **260** | 4.2 | 379 |
| | OpenVLA-OFT (Kim et al., 2025) | **5.0** | 559 | **20.0** | 430 | **55.0** | 480 | **26.7** | 489 |

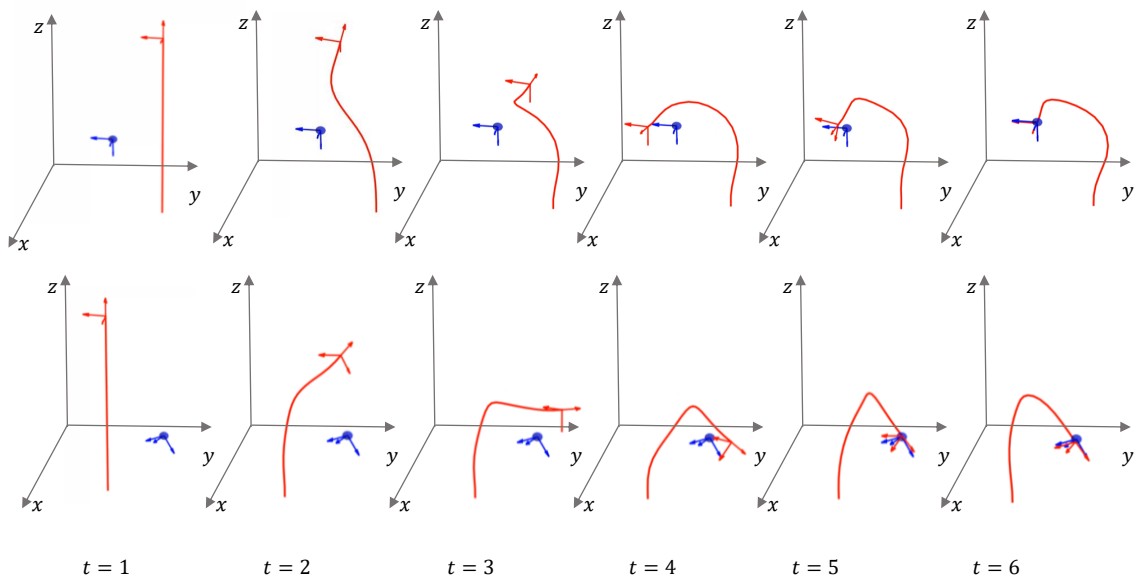

*Figure 10.* Visualization of the trained executor controlling the soft robotic arm to move to the target pose.

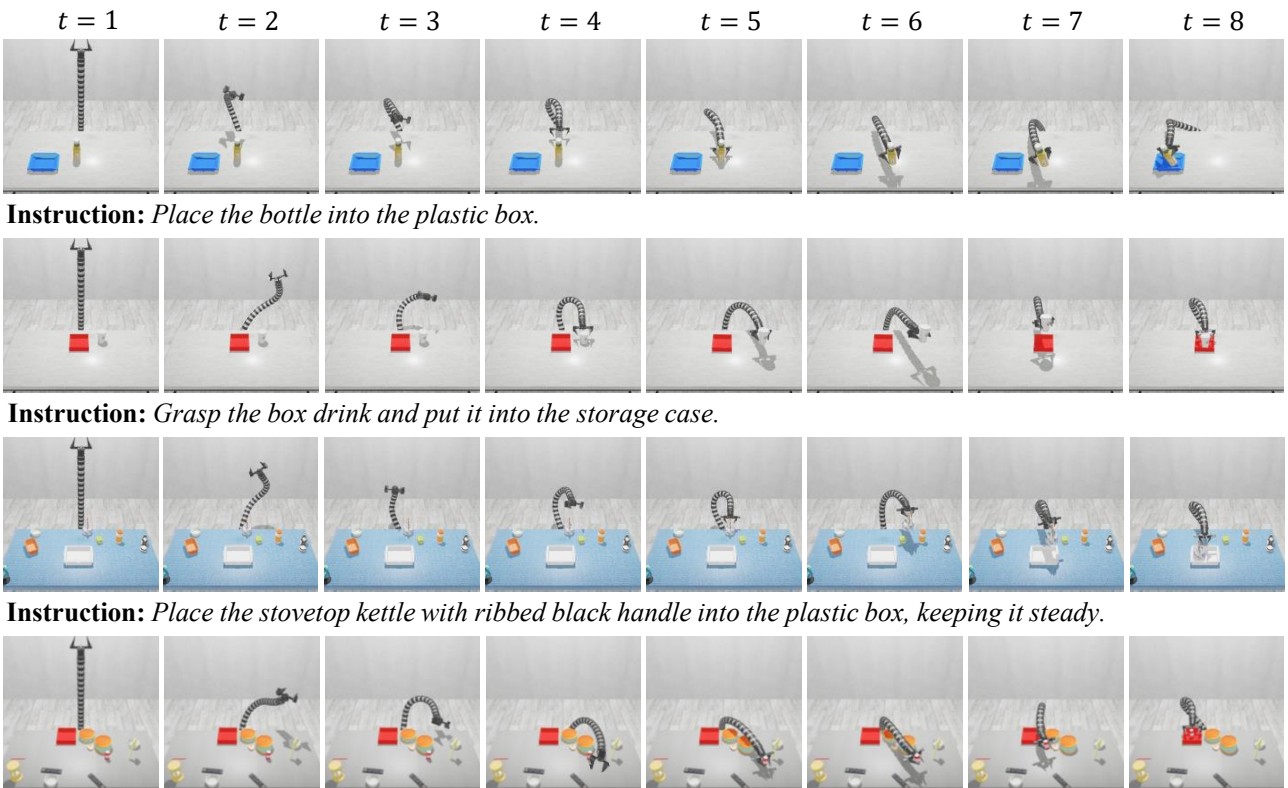

**Instruction:** *Place the bottle into the plastic box.*

**Instruction:** *Grasp the box drink and put it into the storage case.*

**Instruction:** *Place the stovetop kettle with ribbed black handle into the plastic box, keeping it steady.*

**Instruction:** *Take hold of the red soda can with green accents and place it in the storage box.*

*Figure 11.* **Visualization of COLL Task.** The first two are for the clean setting, and the last two are for the randomized setting.

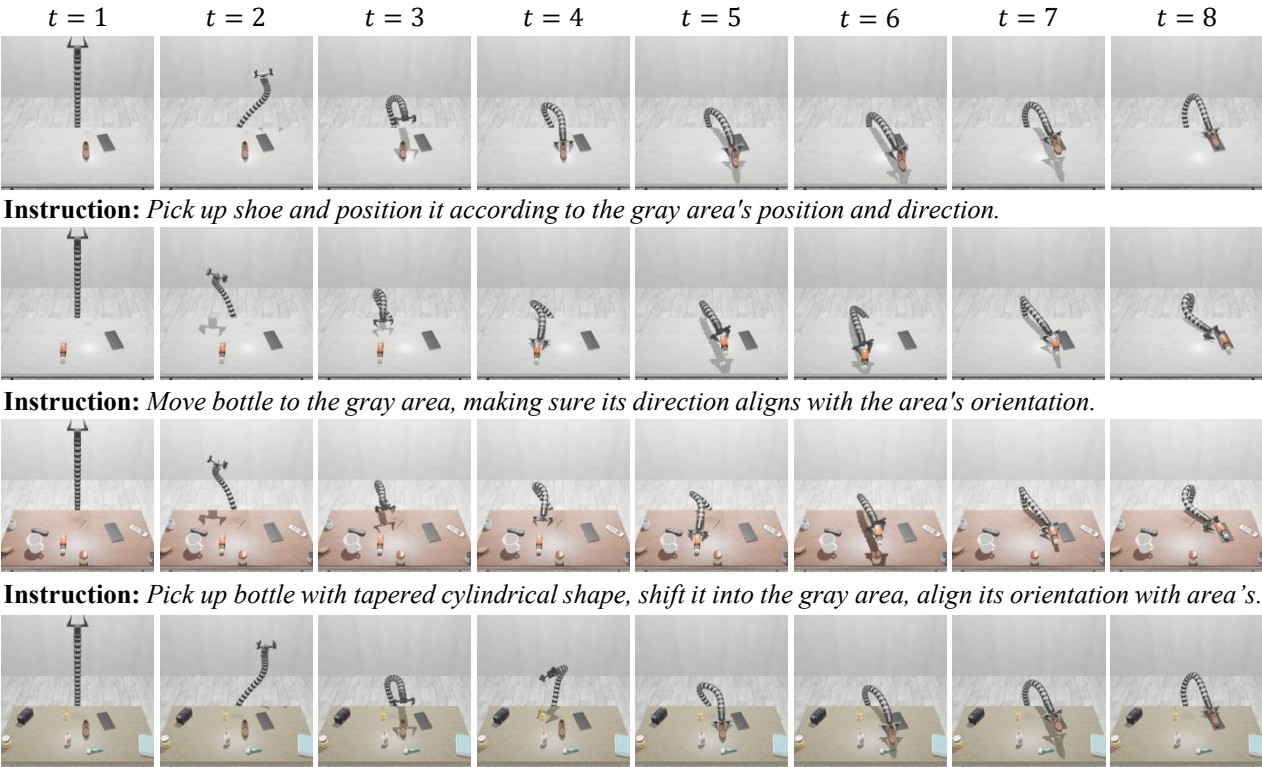

**Instruction:** *Pick up shoe and position it according to the gray area's position and direction.*

**Instruction:** *Move bottle to the gray area, making sure its direction aligns with the area's orientation.*

**Instruction:** *Pick up bottle with tapered cylindrical shape, shift it into the gray area, align its orientation with area's.*

**Instruction:** *Move the brown sports shoe to the gray area, keeping its direction in line with the area's orientation.*

*Figure 12.* **Visualization of ALN Task.** The first two are for the clean setting, and the last two are for the randomized setting.

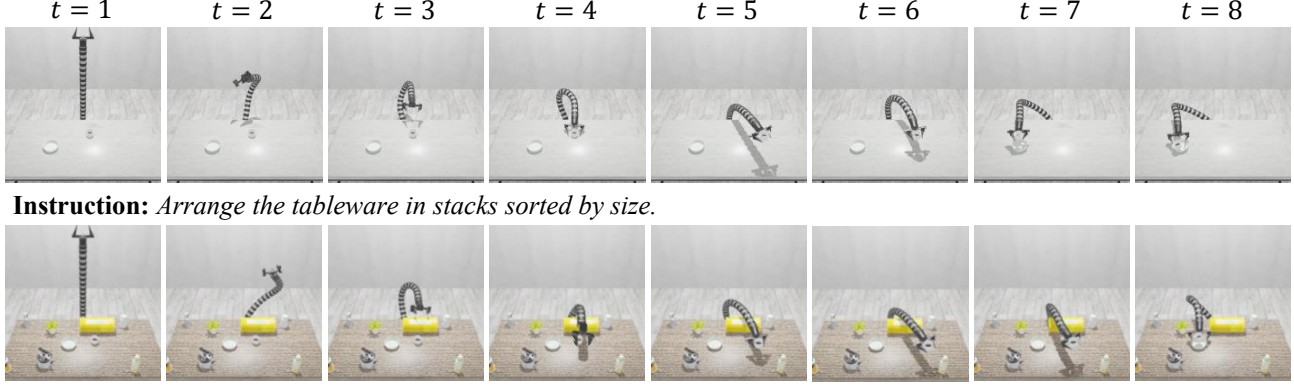

**Instruction:** *Arrange the tableware in stacks sorted by size.*

**Instruction:** *Pile up the eating utensils by size.*

*Figure 13.* **Visualization of STK Task.** The first one is for the clean setting, and the last one is for the randomized setting.

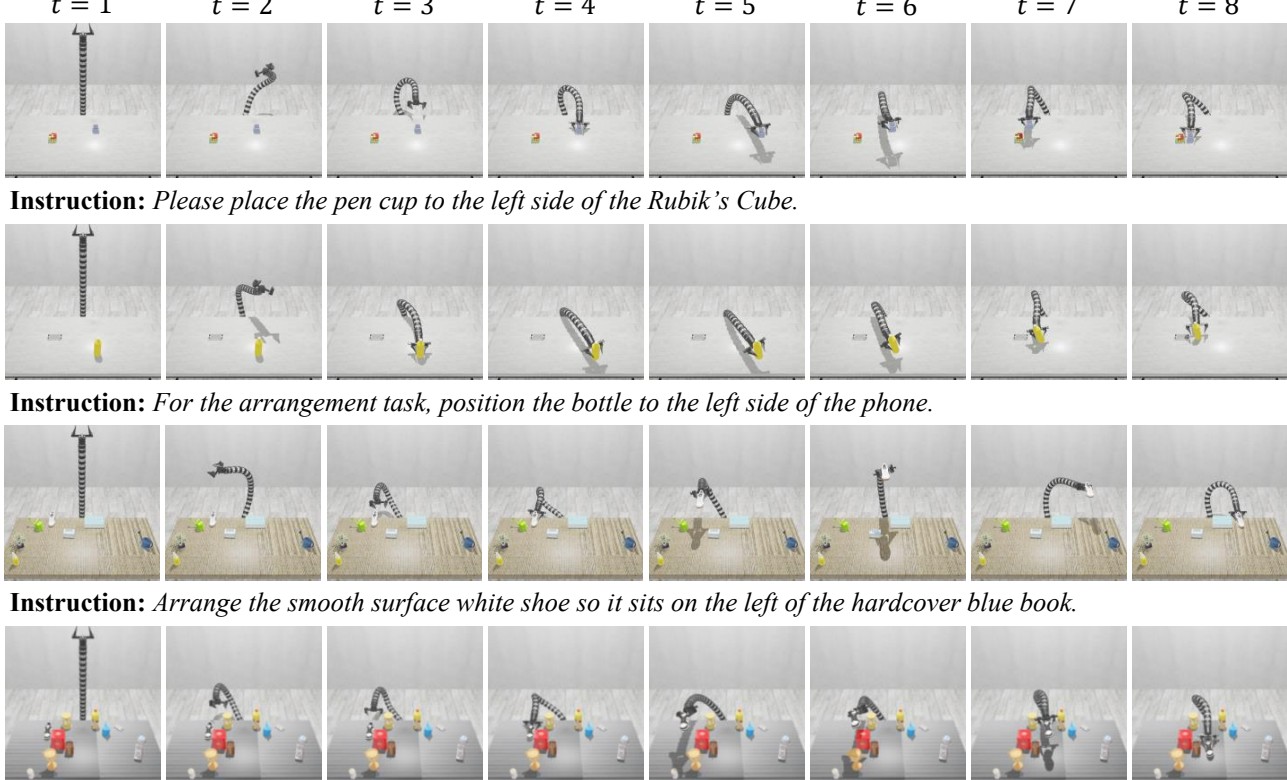

**Instruction:** *Please place the pen cup to the left side of the Rubik's Cube.*

**Instruction:** *For the arrangement task, position the bottle to the left side of the phone.*

**Instruction:** *Arrange the smooth surface white shoe so it sits on the left of the hardcover blue book.*

**Instruction:** *To complete the setup, put the white shoe on the left of the rectangular pencup with rough patterns.*

*Figure 14.* **Visualization of ARR Task.** The first two are for the clean setting, and the last two are for the randomized setting.

