# OpenReview forum: "ManiSoft: Towards Vision-Language Manipulation for Soft Continuum Robotics"
_ICML.cc/2026/Conference — ICML 2026 regular_

### Official Review · Reviewer_h3PD · 2026-02-16

**Soundness:** 2
**Presentation:** 3
**Significance:** 2
**Originality:** 3
**Overall Recommendation:** 4
**Confidence:** 3

**Summary:**

The paper proposes a vision-language-action benchmark, Manisoft, for soft-arm manipulation. The benchmark includes a simulator with realistic soft-body dynamics and contact-rich interactions, and defines four types of tasks. The authors collect 6,300 scene-trajectory pairs and evaluate DP, RDT, and OpenVLA-OFT across different tasks.

**Compliance With Llm Reviewing Policy:**

Affirmed.

**Final Justification:**

The rebuttal address most of my concerns. However, as a benchmark paper, the comparison among only four baselines remains relatively limited. Also, the discussion of future directions is high-level without preliminary results to support it, which limits the potential for real-world applications. But overall, the paper is plausible and makes essential contributions that warrant acceptance. Therefore, I will keep my score in support of it.

**Key Questions For Authors:**

1. See weakness 1 & 2. What is the performance of $\pi$ baselines and/or any soft-arm-specific policies? Can the authors provide concrete research directions or insights for follow-up work beyond benchmarking existing methods?
2. See weakness 3. Can the authors decouple the sources of variation and draw clearer conclusions about the performance drop?
3. See weakness 4. Can the authors elaborate on the evaluation protocol?

**Limitations:**

Yes.

**Strengths And Weaknesses:**

### Strengths
1. The motivation is clear and meaningful. Soft robotic arms can execute policies that are infeasible for rigid arms, while current policies may not fit these scenarios well. The authors build an evaluation suite to verify this and point out several potential limitations.
2. The engineering work is solid, including scene generation and expert-trajectory generation. It can provide a strong benchmark for follow-up work.
3. The authors provide detailed failure-mode analyses.

### Weaknesses
1. The paper primarily presents a benchmark; however, the set of baselines is limited. Only DP, RDT, and OpenVLA-OFT are included in the evaluation, and some commonly used baselines such as $\pi_0$ or $\pi_{0.5}$ are not considered.
2. The benchmark mainly focuses on evaluating existing VLA models. It would be more informative if the authors also evaluated at least one soft-arm-specific policy (even a simplified one) or discussed other plausible solution directions. This could help identify promising research directions for soft-arm tasks, rather than only validating the failure of existing methods.
3. Given the limited baselines and evaluations, some conclusions may be overstated. For example, in Line 357, the paper claims: "This performance gap is likely attributable to differences in model capacity...DP and OpenVLA-OFT each have around 400M parameters." This seems potentially incorrect, since OpenVLA is commonly described as a ~7B-parameter model (and OpenVLA-OFT is an adaptation on top of that). More importantly, it is difficult to attribute the gap to model capacity with only three models. Also, in Line 379, the paper states: "the introduction of obstacles, together with variations in language instructions and scene configurations, substantially increases task difficulty..." Since multiple factors change simultaneously, it is hard to isolate the root cause. It would be beneficial to decouple these factors and report controlled results.
4. As a benchmark paper, the evaluation protocol should be specified in more detail. For example, the criteria for success/failure, stopping conditions, and whether early stopping is used.

---

> ### Author Rebuttal · Authors · 2026-03-31
>
> We are grateful that reviewers appreciate the effectiveness, novelty and impact and comprehensive experiments of our work. We sincerely thank the efforts of the reviewer and will revise our paper accordingly.
>
> # 1. Performance of pi0
> We conducted experiments with pi0 in the clean setting, as shown in the table. Compared with the three baseline methods, the average success rate of pi0 is significantly lower than DP (31.6%) and OpenVLA-OFT (30.4%). It is only slightly higher than RDT (9.2%). Its inference step efficiency is basically similar to the baselines (DP: 520, RDT: 496, OpenVLA-OFT: 527). This shows that pi0 still has clear weaknesses in completing soft robot manipulation tasks. Its overall performance is at a relatively low level among the baseline methods. It also reflects that this method still has substantial room for improvement in the basic policy effectiveness for soft robot VLA tasks.
>
> ||COLL||ALN||STK||ARR||Average||
> |---|---|---|---|---|---|---|---|---|---|---|
> ||success rate|steps|success rate|steps|success rate|steps|success rate|steps|success rate|steps|
> |pi0|8.5|533|13.3|445|15.0|526|7.5|672|11.1|544|
>
> # 2. Soft-Arm Specific Policies
> Existing soft-arm policies primarily target low-level control (e.g., end-effector pose control) and rely on proprioceptive inputs. Their problem formulation differs fundamentally from the VLA setting, which requires joint vision-language reasoning, perception-driven state estimation, and task-level decision-making. As a result, these methods are not directly applicable as baselines for ManiSoft. We have discussed representative soft-robot methods in Section 2 to clarify this distinction.
>
> That said, we do leverage such policies during data generation. In this stage, policies can access richer state information, enabling the generation of stable and high-quality trajectories.
>
> Evaluating representative VLA baselines reveals several consistent failure modes, including inaccurate estimation of the deformable body state, unstable grasping, and frequent collisions. These observations suggest several promising directions:
>
> 1. integrating visual inputs with latent or learned proprioceptive representations;
> 2. incorporating explicit priors over soft-body geometry and dynamics;
> 3. developing hierarchical VLA policies that decouple high-level planning from low-level deformation;
> 4. etc.
>
> We hope ManiSoft can serve as a useful testbed to support and encourage future research along these directions.
>
> # 3. Number of Model Parameters
> We sincerely thank the reviewer for the careful reading and valuable comment. We apologize for the unclear wording in the manuscript. The number reported here refers to the number of trainable parameters rather than the total number of model parameters: OpenVLA-OFT is fine-tuned with LoRA, and the actual number of trainable parameters is approximately 400M. We will revise the manuscript to make this distinction explicit and avoid ambiguity.
> # 4. Ablation Study on Scene Randomization
> |              | ALN  | STK  |
> |---------|------|------|
> | clean | 18.3 | 15.0 |
> | +instruction  | 18.3 | 15.0 |
> | +texture     | 13.3 | 10.0 |
> | +obstacles (randomized) | 1.7  | 2.5  |
>
> We conduct an ablation study to analyze the diversity factors introduced by randomized scenes compared with clean scenes. Using the DP algorithm, we perform factor-wise ablations on the ALN and STK tasks, examining language diversity, scene diversity, and obstacle interference separately. Results show that language diversity alone had little effect on success rates. Adding texture diversity caused a slight 5% drop in both tasks. Further introducing obstacle interference led to a significant performance decline (11.6% for ALN and 7.5% for STK). In summary, language diversity has negligible impact, visual variations cause mild degradation, and obstacle interference is the main reason for severe performance loss.
> # 5. Details on the Evaluation Protocol
> We provide a detailed description of the ManiSoft evaluation protocol. Success is determined using object bounding boxes with task-specific criteria: for COLL, the target object must be placed completely inside the container; for STK, objects must be stably stacked vertically in descending order of size; for ALN, both position and orientation must satisfy predefined error thresholds; for ARR, objects must be arranged in the specified order and roughly aligned along a straight line.
>
> An episode terminates when the task is completed, the simulation crashes due to abnormal manipulation, an object is knocked off the table, or the maximum of 1,000 steps is reached. We apply early stopping to terminate the simulation immediately upon meeting any of these conditions, saving unnecessary computation.
>
> As described in Section 3.2, we use two main metrics: (1) success rate, the proportion of test tasks completed within 1,000 steps; and (2) average completion steps, the average number of inference steps required for successful trials.

---

> > ### Author Rebuttal · Reviewer_h3PD · 2026-04-03
> >
> > Thanks for the authors' detailed responses, which address most of my concerns. However, as a benchmark paper, the comparison among only four baselines remains relatively limited. Also, the discussion of future directions is high-level without preliminary results to support it, which limits the potential for real-world applications. Therefore, I will keep my score.

---

> > > ### Author Response · Authors · 2026-04-07
> > >
> > > We sincerely thank the reviewer for the continued effort and for the recognition of our work. As a pioneering simulator and dataset in the soft robot VLA field, we evaluate several representative VL manipulation methods to provide preliminary insights into soft VL manipulation. Based on these experimental findings, we will conduct more targeted experiments in our future work.

---

### Official Review · Reviewer_vhYS · 2026-03-06

**Soundness:** 3
**Presentation:** 3
**Significance:** 4
**Originality:** 4
**Overall Recommendation:** 5
**Confidence:** 5

**Summary:**

This article introduces ManiSoft, which is the first comprehensive benchmarking tool made specifically for visual-language operations with soft robotic arms. It has three core designs in total: first, a hybrid simulator that combines Elastica (for soft dynamics) and MuJoCo (for rich contact interactions), the two are connected via elastic force constraints, and only RGB observation results are provided to fit the limitations of real-world perception; second, four progressively guided language-based operation tasks, each has both clean and random settings, which are used for basic ability and generalization evaluation respectively; third, an automated hierarchical pipeline, it decomposes tasks into 6-degree-of-freedom path points through a high-level planner, and converts them into torque commands via a low-level reinforcement learning actuator, finally generating 6300 scenario-trajectory pairs and matching corresponding language instructions for them. The authors did benchmark tests on three representative visual-language-action models with this tool, and the results showed the models performed well in clean scenes but their performance dropped sharply in random scenes. This also revealed two core bottlenecks: one is the inaccurate estimation of the proprioceptive state of soft robotic arms based on vision, the other is the failure to fully utilize the arm’s deformability for adaptive obstacle avoidance.

**Compliance With Llm Reviewing Policy:**

Affirmed.

**Key Questions For Authors:**

1. All experiments in this paper are conducted purely in a simulation environment, with no real-world validation. Do you have plans to verify the ManiSoft benchmark and tested models on a physical soft robotic arm? What do you see as the core challenges for the sim-to-real transfer of this benchmark?

2. The current benchmark only supports a single torque-driven soft arm, and does not cover other mainstream actuation types like pneumatic or tendon-driven soft arms. Do you plan to extend the ManiSoft benchmark to support these common soft arm actuation types? Will such an extension require major changes to your core simulator design and trajectory generation pipeline?

3. The analysis of model failure modes in the paper is mainly descriptive, with no quantitative ablation studies on the root causes. Do you plan to carry out quantitative ablation experiments?

**Limitations:**

All experiments were conducted in a simulated environment, and no verification was done on actual physical soft robotic arms. Moreover, the sim-to-real transfer performance of the benchmark and the tested models hasn’t been confirmed yet. This benchmark only supports a single torque-driven soft arm, with poor generalizability. All designed tasks are limited to static desktop operation scenarios, and don’t include more complex settings such as dynamic environments and long-term multi-step tasks. The analysis of model failure modes is mainly descriptive, and the authors don’t conduct quantitative ablation studies on the fundamental causes of these failures. In addition, the authors didn’t fully discuss the limitations of their research and the possible negative social impacts it may bring.
The authors could add a dedicated and independent "Limitations" section in the paper. In this section, they can systematically list and elaborate on the above core limitations, and clarify the objective research constraints that led to these limitations in the current work. They can also add detailed future work plans to address the key technical limitations mentioned above. Finally, it’s necessary to include a special discussion on the potential negative social impacts of this work.

**Strengths And Weaknesses:**

Soundness
Strengths: The technical design is rigorous and methodologically appropriate. The experiments are well-designed, especially for the setting of core parameters. All the research claims in the paper are fully supported by experimental results.
Weaknesses: The failure mode analysis is only descriptive, and lacks quantitative ablation studies. There is no verification for the sim-to-real transfer performance of both the benchmark and the tested models. In addition, the benchmark only supports a single type of torque-driven soft arm.

Presentation
Strengths: This paper is clearly written and has a logical structure overall. Sufficient implementation details are also provided in the main text and appendix for reference.
Weaknesses: The analysis of model failure modes is mainly descriptive. The authors don’t conduct quantitative ablation studies to explore the root causes of these failures.

Significance
Strengths: This paper addresses an important research gap in the field of soft robotic arm vision-language manipulation. The standardized testbed and large-scale dataset it provides can strongly support future related research. What’s more, the core bottlenecks identified in the paper offer clear and valuable directions for subsequent studies.
Weaknesses: The benchmark only supports torque-driven soft arms, which limits its practical application scope. The designed tasks are restricted to static tabletop scenarios, and fail to cover complex real-world use cases. Besides, the coverage of baseline models is limited, which reduces the persuasiveness of the benchmark’s evaluation results.

Originality
Strengths: This paper is the first work to build a complete benchmark for soft robotic arm vision-language manipulation. Its hierarchical trajectory pipeline also innovatively solves the data generation problem in high-dimensional torque control.
Weaknesses: The paper doesn’t propose a completely new VLA model for the specific characteristics of soft arms. Its novelty is mainly reflected in benchmark construction, and there is no breakthrough innovation in core algorithms.

---

> ### Author Rebuttal · Authors · 2026-03-31
>
> Thank you for your thoughtful feedback and the recognition of our work. Below we address specific questions.
>
> # 1. Sim-to-Real Discussions on Soft Robotic Arm
> Robot VLA is a complex long-term challenge involving data collection, policy training, simulation, sim-to-real transfer, and real-robot deployment. This work presents the first dedicated simulation benchmark, environment, and training dataset for vision-language-based soft robot manipulation, addressing the current gap in soft robot VLA platforms and data.
>
> Sim-to-real transfer for soft arms is significantly harder than for rigid arms. Beyond typical perception and policy gaps, soft robots suffer from large continuous deformation, lower end-effector precision, severe contact deformation, and immature data collection. Teleoperation, calibration, state estimation, and safe interaction also demand substantial engineering effort. We believe these challenges require broader community collaboration beyond a single benchmark study.
>
> To this end, we are currently building a real-world soft robot data collection pipeline and flywheel. Our present hardware platform is a pneumatic-driven soft robotic arm with five segments, each containing three bellows, resulting in 10 independent air channels (see demo.mp4 [1]). We hope that the ManiSoft simulation environment can serve as an open **pioneering infrastructure** to support and facilitate future community research in soft robot VLA.
>
> [1] https://anonymous.4open.science/r/Manisoft
> # 2. Extension to Common Actuation Types
> Our simulator relies on Cosserat rod theory, abstracting all actuation as generalized internal/external loads (distributed forces and torques along the rod), enabling a unified, mechanism-agnostic actuation representation [1].
>
> In our implementation, control is applied directly as torques, making our approach independent of specific actuation types and focused on mechanical outcomes. We acknowledge real-world performance may differ due to actuator constraints and will clarify this limitation in the revision.
>
> Since the Cosserat rod formulation is driven by forces and torques, incorporating actuator-level models amounts to introducing a mapping from actuator variables to these quantities. This can be achieved without modifying the core simulator or the trajectory generation pipeline, and we will explore this extension in future work.
>
> [1] Naughton N, et al. Elastica: A compliant mechanics environment for soft robotic control. IEEE RAL, 2021.
> # 3. Task Scenario Design
> The current ManiSoft benchmark focuses primarily on static desktop scenes. This design choice aligns with the common practice in embodied AI and robotic manipulation research. Many established vision-language manipulation benchmarks for rigid arms, such as RoboTwin, LIBERO, and CALVIN, also mainly use static desktop environments. These scenes effectively cover practical daily tasks, serving as a solid foundation for studying vision-language-action models.
>
> We agree that more complex settings, such as dynamic environments and long-horizon multi-step operations, are valuable directions for future exploration. In our future work, we plan to gradually introduce more diverse tasks and environments, including dynamic scenes and longer-horizon operations.
> # 4. Statistical Analysis of Failure Modes
>
> || Colliding with Objects | Stopping Movement | Large Swings | Recognition Errors | Unstable Grasping | Obstacle Avoidance Failure |
> |:---:|:---:|:---:|:---:|:---:|:---:|:---:|
> |DP|69%|0%|5%|16%|2%|7%|
> |OpenVLA-OFT|35%|27%|7%|15%|6%|11%|
> |RDT|21%|0%|1%|74%|4%|0%|
>
> We conducted a statistical analysis of the failure modes for all baseline methods. The results are shown in the table above. The dominant failure cause for the DP model is "colliding with objects" (69 %), while the RDT model is overwhelmingly limited by “recognition errors” (74 %). In contrast, the failure modes of OpenVLA-OFT are more evenly distributed, with the two primary issues being “colliding with objects” (35 %) and “stopping movement” (27 %).
> # 5. Limitations and Social Impacts
> We will add a dedicated “Limitations” section discussing both the core limitations of our work and potential negative societal impacts.
>
> We will address key limitations, including the current lack of support for pneumatic or tendon-driven actuation and the restriction to static desktop environments (without dynamic objects or long-horizon tasks). We will also outline future work plans, such as extending support for diverse actuation types and soft arm morphologies, improving sim-to-real consistency, conducting hardware validation, and creating more complex dynamic scenes.
>
> Additionally, we will discuss societal impacts: positively, ManiSoft aims to advance safe and friendly soft robotics applications in flexible interaction, service, and medical assistance; on the negative side, we will objectively note potential risks related to safety and privacy if the technology is misused.

---

> > ### Author Rebuttal · Reviewer_vhYS · 2026-04-03
> >
> > I sincerely appreciate the authors’ thorough, thoughtful response to all the concerns raised in my review, as well as the targeted supplementary work completed to address these points.
> > The authors have rigorously and fully responded to every core question: their clarification of sim-to-real challenges and ongoing real-world hardware work provides a clear roadmap for the benchmark’s future extension; the technical explanation of supporting diverse actuation types based on Cosserat rod theory demonstrates the inherent scalability of the core design; the supplementary quantitative statistical analysis of failure modes directly fills the gap in the original descriptive analysis; and the commitment to add a dedicated Limitations section with systematic discussion of research constraints and social impacts further improves the manuscript’s completeness and academic standard.
> > I fully recognize and highly commend the authors’ rigorous academic attitude and the meaningful improvements to this work. All my key concerns have been properly addressed, and the supplementary content has notably enhanced the quality and practical value of the manuscript. I maintain my original recommendation of acceptance for this submission.

---

> > > ### Author Response · Authors · 2026-04-07
> > >
> > > We appreciate your acknowledgment of our rebuttal and your positive feedback. We are pleased to have your concerns addressed and sincerely appreciate your thorough review and insighful comments.

---

### Official Review · Reviewer_RqAU · 2026-03-11

**Soundness:** 3
**Presentation:** 3
**Significance:** 3
**Originality:** 2
**Overall Recommendation:** 4
**Confidence:** 3

**Summary:**

This paper introduces ManiSoft, a benchmark for enhancing vision-language manipulation in soft robotic arms. To tackle the challenge of simulating the dynamics of soft arms, the authors developed a hybrid simulator that combines elastic dynamics with contact-rich interactions. Using this simulation environment, they created an automated pipeline for generating task trajectories. In total, they collected over 6K diverse demonstrations of tabletop tasks. The authors systematically evaluate a range of representative policy learning algorithms for vision-language manipulation, analyzing their strengths and weaknesses across different tasks.

**Compliance With Llm Reviewing Policy:**

Affirmed.

**Final Justification:**

I appreciate the authors' feedback on addressing the sim-to-real gap in soft robots. My concerns regarding the inconsistencies in the action space and the transfer of vision to proprioception have largely been resolved by the supplemented visual results. Therefore, I am pleased to increase my recommendation score.

**Key Questions For Authors:**

1.	A key premise of the work is that soft arms perform better than rigid arms when it comes to reaching around obstacles. However, many of the tasks demonstrated could also potentially be accomplished by a rigid arm with an extended workspace, such as one mounted on a lifting platform or a linear rail. Did the authors consider including such a baseline in their evaluation? This would help determine whether the advantages of ManiSoft’s tasks arise from continuous deformation rather than simply increased reach or alternative kinematics.
2.	All experiments presented in the paper are conducted in a simulation environment. Could the authors provide some preliminary real-world validation to verify the effectiveness of the policy trained in this simulation when applied to real-world scenarios? If this is not possible, could the authors discuss the expected sim-to-real gap specifically for soft robotic arms, considering that factors like material fatigue and non-linear elastic behaviors are challenging to model?
3.	The authors specify the use of Young's modulus and Poisson's ratio for the soft arm in Table 4. Could they clarify the reasoning behind selecting these particular values? How sensitive are the benchmark results to variations in these parameters? Additionally, how can we obtain these values in a real-world setting?
4.	The paper mentions that scenes are generated by randomly sampling and placing objects within the workspace. Did the authors implement a geometric or physical constraint during this sampling to ensure scenes are physically plausible (e.g., no interpenetrating geometries) before the trajectory generation begins? In addition, there is a lack of discussion on related work [a-c] concerning scene generation for robotic tasks.
5.	The paper does not specify the details of the rules used in the rule-based planner for each task, and it remains unclear how robust these rules are across the 6,300 diverse scenes. Could the authors provide more details on this?
6.	What is the time required for the method to generate a successful task demonstration on average (including both scene and trajectory generation)?

References:

[a] Architect: Generating Vivid and Interactive 3D Scenes with Hierarchical 2D Inpainting. NeurIPS 2024.

[b] TabletopGen: Instance-Level Interactive 3D Tabletop Scene Generation from Text or Single Image. arXiv 2025.

[c] GAIA: Generating Task Instruction Aware Simulation Grounded in Real Contexts Using Vision-Language Models. Robotics and Automation Letters 2025.

**Limitations:**

It may be difficult to obtain a soft robot rather than a rigid robot in the real world.

**Strengths And Weaknesses:**

**Strengths**:

-	The paper presents an automatic trajectory generation pipeline that can generate diverse demonstrations for policy learning.
-	The authors developed a hybrid simulator coupling elastic dynamics with contact-rich interactions to model the dynamics of soft robotic arms.

**Weaknesses**:

-	The need for soft arms in solving the proposed benchmark tasks requires further justification.
-	A lack of real-world results undermines the practicality of the proposed framework.
-	Missing implementation details in scene and trajectory generation.

---

> ### Author Rebuttal · Authors · 2026-03-31
>
> We sincerely thank the efforts of the reviewer and will revise our paper accordingly.
> # 1. Advantages of Soft Arms over Rigid Arms
> While rigid arms rely on external setups to handle occluded tasks, soft arms can naturally achieve this via continuous body deformation without extra hardware. Despite such unique merits, VLA research for soft robots remains scarce. We thus propose ManiSoft, the first dedicated benchmark for soft-robotic vision-language-action tasks.
>
> As a pioneering benchmark, ManiSoft includes diverse tasks from basic manipulation to spatial reasoning, fully exploiting soft arms’ compliance and deformability. Many scenarios feature obstacles between the arm and target, requiring flexible bypassing. As demonstrated in fig1.png [1], the soft arm deforms its body to grasp an occluded bottle successfully, while a Franka rigid arm fails due to joint and workspace limits (fig2–3.png [1]).
>
> [1] https://anonymous.4open.science/r/Manisoft
>
> # 2. About Real-World Validation and Sim-to-Real Gap
> Robot vision-language manipulation is a complex long-term challenge. Instead of solving all existing issues, our core contribution is to fill a critical gap by building the first dedicated simulation benchmark, environment, and dataset for soft robot vision-language manipulation.
>
> In VLA research, “simulation pre-training + real-world fine-tuning” is widely adopted [1]. ManiSoft provides high-quality data and a standardized platform, enabling low-cost large-scale expert demonstration collection impractical on real hardware. 、
>
> Soft robots face unique sim-to-real and real-world challenges: large continuous deformation reduces positioning accuracy; perception noise, contact deformation, and interference are severe; and data collection methods remain immature. Calibration, state estimation, and safe interaction also require heavy engineering, making VLA deployment much harder than on rigid arms. These challenges demand broader community collaboration.
>
> We are developing a real-world soft-robot data pipeline and data-flywheel infrastructure as key future work. Our hardware is a 5-segment pneumatic soft arm with 10 independent control channels (demo.mp4 [2]). We hope ManiSoft will serve as an open infrastructure for future soft robot VLA research.
>
> [1] Nvidia J B, et al. Gr00t n1: An open foundation model for generalist humanoid robots. 2025.
>
> [2] https://anonymous.4open.science/r/Manisoft
>
> # 3. Selection of Soft Arm Properties
> We chose the values of Young's modulus and Poisson's ratio following the parameter settings in Elastica. This parameter combination matches the typical stiffness characteristics of soft robotic arms and ensures that the simulation environment aligns with the physical properties of real hardware [1].
>
> In practice, these parameters can be directly measured via standard material tests. Future users can flexibly adjust these values to better match their own hardware platform.
>
> [1] Nvidia J B, et al. Gr00t n1: An open foundation model for generalist humanoid robots. 2025.
>
> # 4. Related Works on Scene Generation
> We agree with the reviewer that works [a–c] make important contributions to high-fidelity scene generation.
> Our work instead focuses on a soft-robot vision-language manipulation benchmark. We use random object placement to boost data diversity and generalization, with geometric collision checks to avoid robot–obstacle conflicts, following standard practices in rigid-arm benchmarks [1,2].
>
> [1] Chen T, et al. Robotwin 2.0: A scalable data generator and benchmark with strong domain randomization for robust bimanual robotic manipulation. arXiv preprint, 2025.
>
> [2] Liu B, et al. Libero: Benchmarking knowledge transfer for lifelong robot learning. Advances in Neural Information Processing Systems, 2023.
>
> # 5. Details for High-Level Planning
>
> The high-level rules are tailored to each task. Take the COLL task as an example, they consist of seven steps: move to pre-grasp pose, move to grasp pose, grasp the object, set torque to zero and lift using the soft arm’s elasticity for 5 seconds, move 0.1 m above the storage box, move down 0.1 m, and release.
>
> For each object, both grasp and pre-grasp poses are pre-calibrated and stored in the asset library. Grasp poses are manually annotated to suit the soft arm, while the pre-grasp pose is placed 5–15 cm behind it. This two-step approach significantly reduces the risk of knocking over the object, improving data generation efficiency and stability.
>
> # 6. Time Required for Data Generation
> The average time required to obtain one successful trajectory is about 100 seconds. Afterwards, each trajectory is rendered 3 times (clean ×1 and randomized ×2), taking approximately 150 minutes in total. With 80 parallel processes, the total time for generating all 6,300 samples is 66 hours, which corresponds to an average of 38 seconds per sample.

---

> > ### Author Rebuttal · Reviewer_RqAU · 2026-04-01
> >
> > I thank the authors for their response and for providing the hardware demo. While I appreciate the effort required to create a simulation benchmark for soft-robot VLA manipulation, I still have concerns about the sim-to-real gap and the benchmark's overall utility. These concerns remain largely unaddressed for the following reasons:
> > 1. **Morphological Discrepancy**: The simulation model, shown in Figures 1 and 2 in the manuscript, is based on a slender, continuous-looking arm that is modeled using Cosserat rod theory. In contrast, the hardware depicted in the author's response (demo.mp4) consists of a 5-segment pneumatic bellows system. These two designs exhibit fundamentally different bending behaviors, occlusion patterns, and visual characteristics.
> > 2. **Action Space Inconsistency**: The paper's policy predicts external torques $\tau_e$, while the actual hardware has 10 independent pneumatic control channels. This change signifies a major transition from a continuous torque-based action space to a discrete pressure-based one. It remains uncertain how a VLA model pre-trained on the former would offer a meaningful starting point for the latter.
> > 3. **Vision-Proprioception Transfer**: The authors emphasize that "inaccurate visual estimation of proprioceptive state" is a major point of failure in their benchmark. Because there are significant differences in visual appearance between soft robots in simulation and those in the real world, it is highly likely that a vision-based policy trained in ManiSoft will have difficulty generalizing to the real-world pneumatic arm's visual features.
> >
> > If there's no clear way to bridge the gaps between simulation and real-world environments, the benchmark may focus on an ideal model rather than be useful for real-world robotics. I would appreciate it if the authors could clarify whether ManiSoft supports customizable arm morphologies to match specific hardware, such as the one shown in their demo.

---

> > > ### Author Response · Authors · 2026-04-07
> > >
> > > We thank the reviewer for recognizing our efforts in building the simulation benchmark. To address the concerns regarding the sim-to-real gap, we provide additional clarification from the following aspects, supported by two experiments as shown below.
> > >
> > > ### 1. Morphological Discrepancy
> > > **Regarding occlusion patterns and visual characteristics**, we follow [1] to render the appearance. This appearance is similar to that of many soft robots in the community [4][5], resulting in relatively small differences in occlusion patterns and visual characteristics.
> > >
> > > **Regarding bending behaviors**, the Cosserat rod formulation is one of the representative modeling methods in the soft robotics field [2][3]. As shown in **Experiment 1**, using this modeling approach, it is possible to map between simulation and real hardware morphology, achieving consistent bending behavior.
> > >
> > > As shown in **Experiment 2**, we use a model pre-trained in simulation and deploy it on a real 5-segment pneumatic soft arm (a specific platform used in our lab). Despite morphological differences between simulation and hardware, the task can be successfully completed after fine-tuning with a small amount of real-world data.
> > >
> > > [1] Rao P, et al. How to model tendon-driven continuum robots and benchmark modelling performance. Frontiers in Robotics and AI, 2021.
> > >
> > > [2] Tummers M, et al. Cosserat rod modeling of continuum robots from Newtonian and Lagrangian perspectives. IEEE Transactions on Robotics, 2023.
> > >
> > > [3] Li H, et al. Piecewise linear strain Cosserat model for soft slender manipulator. IEEE Transactions on Robotics, 2023.
> > >
> > > [4] https://anonymous.4open.science/r/Manisoft/fig4.jpg
> > >
> > > [5] https://anonymous.4open.science/r/Manisoft/fig5.jpg
> > >
> > > ### 2. Action Space Inconsistency
> > > Torque is an abstract control representation, whereas pneumatic pressure is a physical actuation mechanism. In practical hardware implementations, soft robotic arms may employ different actuation methods, such as pneumatic or tendon-driven actuation. Despite these differences, such actuation methods can be modeled using a torque-based formulation. Therefore, torque control is adopted in our simulator. As demonstrated in **Experiment 1**, actions in simulation are mappable to those on the physical robot.
> > >
> > > From a learning perspective, simulation enables the generation of large amounts of data at low cost, allowing the model to learn vision-language semantics, high-level planning, and interaction experience from diverse manipulation trajectories.
> > > As shown in **Experiment 2**, with a small amount of real-world data for fine-tuning, the task can be successfully completed, demonstrating preliminary transferability from simulation pretraining to real-world experiments.
> > >
> > > ### 3. Vision–Proprioception Transfer
> > > We would like to clarify that the proprioceptive state in our work refers not only to the shape of the soft body (i.e., its curve representation in the world coordinate system), but also includes internal states such as shear stress and bending stress.
> > >
> > > We aim for the model trained in our simulator to learn an internal representation that maps body shape to these internal states. Since this process is independent of visual appearance, it can potentially provide useful priors for real-world experiments. As shown in **Experiment 2**, the model pre-trained on our simulation data achieves preliminary performance after fine-tuning on a real robot.
> > >
> > > In future works, we plan to extend the simulator to support more soft robot morphologies and contribute further to the community.
> > >
> > > ### Experiments
> > > **Experiment 1: Action Space Mapping between Simulator and Hardware**
> > >
> > > We implement a naive mapping between torque control in simulation and pneumatic control in real systems. Specifically, we compute the equilibrium configuration of the soft arm under torque control based on Cosserat rod theory, and then map this configuration to corresponding pneumatic control signals through an inverse statics model.
> > >
> > > We randomly generate four cases for qualitative evaluation. As shown in [6-9], the simulation and real hardware exhibit consistent morphology after mapping.
> > >
> > > [6] https://anonymous.4open.science/r/Manisoft/map1.png
> > >
> > > [7] https://anonymous.4open.science/r/Manisoft/map2.png
> > >
> > > [8] https://anonymous.4open.science/r/Manisoft/map3.png
> > >
> > > [9] https://anonymous.4open.science/r/Manisoft/map4.png
> > >
> > > **Experiment 2: Real-World Policy Fine-Tuning**
> > >
> > > To further demonstrate the feasibility of sim-to-real transfer, we adopt a model pre-trained in simulation and fine-tune it with a small amount of real-world data. Specifically, we use the DP model trained on the COLL task as the pretrained model, collect real-world data, and perform fine-tuning on our real soft robotic arm.
> > >
> > > As shown in [10], the fine-tuned model successfully grasps the small bottle on the left and places it into the box on the right.
> > >
> > > [10] https://anonymous.4open.science/r/Manisoft/real_vla.mp4

---

### Official Review · Reviewer_8hq7 · 2026-03-12

**Soundness:** 4
**Presentation:** 4
**Significance:** 3
**Originality:** 3
**Overall Recommendation:** 4
**Confidence:** 5

**Summary:**

This paper introduces ManiSoft, a benchmark designed to evaluate vision-language manipulation for soft robotic arms, which typically lack reliable proprioception and require complex low-level actuation. The benchmark features a tailored simulator that merges soft-body dynamics with rigid environmental interactions using an elastic force constraint, forming the basis for four distinct tabletop manipulation tasks. To support policy learning, the authors built an automated pipeline that generates 6,300 expert trajectories by combining a high-level rule-based planner with a low-level reinforcement learning controller. Evaluation of baseline vision-language models shows a notable performance drop in randomized scenes, primarily because the models struggle to estimate proprioceptive states from visual inputs and fail to utilize the arm's deformability for obstacle avoidance.

**Compliance With Llm Reviewing Policy:**

Affirmed.

**Final Justification:**

This work fills a gap in soft robotics vision language manipulation with some inspiring insights. Despite some minor concerns I recommend accepting the paper.

**Key Questions For Authors:**

Q1: The paper notes that the low-level reinforcement learning executor only achieves a 54% success rate. How does this high failure rate affect the generation of the 6,300 expert trajectories? Do you just filter out the failed attempts, and is it possible that the "successful" trajectories are still suboptimal and causing the poor baseline performance?

Q2: The simulator connects the soft body and the rigid end-effector using a virtual spring constraint. However, there are no real-world experiments to show that this method matches physical reality. Can you provide any hardware validation or data showing that this virtual spring setup accurately models real soft arm contact dynamics? Some models may be trained with real world data only, if the dynamic gap between your simulation and real world dynamics is too large, these model may have bad performance in the benchmark, while actually doing well enough in real world.

**Limitations:**

Possible limitations include: sim-to-real gap as discussed above on the key questions, data generation efficiency, and the single morphology that was used in the paper - Soft robots have many different actuation methods, such as pneumatic or tendon-driven systems. The authors should note that their baseline results might not apply to all soft robot shapes and designs.

**Strengths And Weaknesses:**

By combining Elastica for soft-body dynamics and MuJoCo for contact-rich interactions using an elastic force constraint, the authors successfully create a stable simulation environment for soft arms. The hierarchical trajectory generation pipeline, which uses a high-level rule-based planner and a low-level reinforcement learning controller, is a robust method for collecting large-scale expert data without requiring manual teleoperation. The authors also provide ablation studies to justify the stability reward used in their RL training.

A major weakness is that the paper lacks real-world experiments to demonstrate the consistency between the simulation and real-world dynamics. Without hardware validation, it remains unproven whether the simulated elastic constraints and visual observations accurately reflect the physical behavior of actual soft arms. Furthermore, the performance of baseline models drops drastically in randomized scenes, indicating that the benchmark might be excessively difficult or that the visual-to-proprioceptive gap is too severe for current models.

Overall the paper is structured clearly and the narrative is easy to follow.

For significance, this work addresses a highly relevant but neglected problem in robotic manipulation. While most current benchmarks focus on rigid arms, soft robotic arms are essential for operating in confined or cluttered spaces where fixed morphologies fail. The approach to solving the simulation bottleneck—coupling a specialized soft-body simulator with a rigid-body physics engine via virtual springs—is a creative and practical combination of existing techniques.

---

> ### Author Rebuttal · Authors · 2026-03-31
>
> Thank you for your meticulous review and constructive comments. Below we address specific questions.
> # 1. Virtual Spring Constraint and Real-World Dynamics
> Unlike rigid manipulators, soft-arm platforms remain highly heterogeneous in morphology, material properties, actuation, and end-effector mounting. As a result, real-world consistency is inherently platform-dependent: validation against one specific soft arm would only establish fidelity to that particular system, rather than provide a general validation for soft-arm manipulation more broadly. Therefore, ManiSoft is intended as a controlled and scalable benchmark/testbed, rather than a fully hardware-calibrated digital twin of a specific platform. Under this scope, it remains useful for comparative evaluation, large-scale data generation, and failure analysis, while capturing key challenges in soft-arm vision-language manipulation such as visual state inference, vision-language-action alignment, and deformable planning.
>
> The connection between the soft body and the rigid end-effector is modeled as a compliant penalty-based coupling. We use this formulation because it is physically grounded, numerically stable, and practically useful for coupling deformable body dynamics with rigid-body interaction. Rather than claiming exact reproduction of real-world soft-arm contact dynamics, we view it as a reasonable approximation for coordinated compliant manipulation.
>
> More broadly, ManiSoft is built on physically grounded dynamics to provide a controlled and reproducible environment for soft-arm manipulation research. Strong real-world performance will still require platform-specific calibration or adaptation. In this sense, ManiSoft is not meant to replace hardware-specific tuning, but to enable lower-cost data generation, rapid algorithm validation, and comparative evaluation before hardware deployment.
>
> [1] Todorov E, et al. Mujoco: A physics engine for model-based control. IEEE/RSJ international conference on intelligent robots and systems. 2012.
>
> [2] SOFA, Simulation Open-Framework Architecture (c) 2006 INRIA, USTL, UJF, CNRS, MGH
>
> # 2. RL Executor for Trajectory Generation
> We trained the executor via reinforcement learning (RL). For end-effector pose control, it achieves an average position error of 0.048 (4.8% of the soft arm length) and an average orientation error of 0.305, showing reasonable precision. As average errors poorly reflect trajectory success, we define success as position error < 0.03 and orientation error < 0.3 (Appendix C), thresholds that generally ensure successful grasping. Failed samples are discarded during trajectory generation, and we manually filtered out instances with excessive deformation, collisions, or unsmooth motion. This results in 6,300 high-quality stable expert trajectories to reduce training noise.
> # 3. Data Generation Efficiency
> Generating one trajectory takes 10 seconds. Considering the success rate, the average time to generate a successful trajectory is 100 seconds. Each successful trajectory is rendered three times (1 clean + 2 randomized), with each rendering taking ~50 minutes, for a total of ~150 minutes per trajectory. Since trajectory generation accounts for only 1.1% of the total data generation time, its success rate has negligible impact on overall efficiency. Using 80 parallel processes, the complete data generation took 66 hours.
> # 4. Dynamics Modeling of Soft Robotics
> Our simulator does not model specific actuators such as pneumatic chambers or tendons. Instead, it uses Cosserat rod theory, representing actuation as generalized internal/external loads (distributed forces and torques along the rod), which provides a unified representation across actuation modes [1].
>
> Control in our simulator is applied directly as torques, independent of any physical actuation mechanism. We acknowledge that real-world performance may vary with how physical actuators realize these torques, and we will clarify this limitation in the revision. Future work will integrate actuator-specific models to address this gap.
>
> [1] Naughton N, et al. Elastica: A compliant mechanics environment for soft robotic control. IEEE RAL, 2021.
>
> # 5. Performance of Baseline Models in Randomized Scenes
> ||ALN|STK|
> |---|---|---|
> |clean|18.3|15.0|
> |+instruction|18.3|15.0|
> |+texture|13.3|10.0|
> | +obstacles (randomized)|1.7|2.5|
>
> To investigate the causes of model performance degradation in randomized scenes, we performed ablation studies on three diversity factors: language instruction diversity, visual diversity, and obstacle interference. Experiments were conducted on the ALN and STK tasks using the DP algorithm.
>
> Results show that language diversity alone had little effect on success rates. Adding texture diversity caused a slight 5% drop in both tasks. Further introducing obstacle interference led to a significant performance decline (11.6% for ALN and 7.5% for STK).

---

> > ### Author Rebuttal · Reviewer_8hq7 · 2026-04-03
> >
> > Most of my concerns have been resolved. I will keep my rating positive.

---

> > > ### Author Response · Authors · 2026-04-07
> > >
> > > We are glad to have addressed your concerns. We sincerely thank you for your positive feedback and recognition of our work.

---

### Decision · Program_Chairs · 2026-04-30

**Decision:**

Accept (regular)

**Comment:**

This paper introduces ManiSoft, a benchmark for vision-language manipulation with soft robotic arms. The review outcome is overall positive and became somewhat stronger after rebuttal.

Reviewers broadly agreed on the paper’s main strengths. Reviewers agree that soft-arm vision-language manipulation is underexplored, and that the benchmark fills the gap by providing a reasonably complete evaluation setup rather than only a simulator or a dataset. Multiple reviewers also viewed the engineering work as substantial. The most persistent issue is the lack of convincing real-world validation. Additionally, the benchmark evaluation remains somewhat narrow, with limited baselines and only high-level discussion of future methods.

Overall, I lean toward acceptance.